EMBO
Molecular Medicine

# Defective chaperone-mediated autophagy in the retinal pigment epithelium of age-related macular degeneration patients

Juan Ignacio Jiménez-Loygorri [1,2✉], Peng Shang[3], Ibrahim Bayramoglu [4], Raquel Gómez-Sintes [1], Adrián Martín-Segura [5,6,14], Helena Ambrosino[3], Johnson Hoang [3], Antonio Díaz [5,6], Zhaohui Geng [7,8], Evripidis Gavathiotis [9,10,11], James R Dutton[7,8], Jörn Dengjel [4], Ana Maria Cuervo [5,6,10], Deborah A Ferrington [3,12✉] & Patricia Boya [1,13✉]

## Abstract

**Autophagy is one of the main intracellular recycling systems and its impairment is considered a primary hallmark of the aging process. Defective macroautophagy in the retinal pigment epithelium (RPE) has been described in age-related macular degeneration (AMD), a blindness-causing disease that affects roughly 200 million patients worldwide. The relevance of chaperone-mediated autophagy (CMA), a selective type of autophagy for proteins containing a KFERQ-like motif, in RPE cell biology and homeostasis remains to be elucidated. Here we describe decreased CMA activity in the RPE of AMD patients compared to healthy age-matched controls, along with accumulation of substrate proteins, and in donor-derived iPSC-RPE cells, which we used to further characterize AMD-associated alterations of cellular homeostasis derived from proteotoxicity. Treatment with CA77.1 (CMA activator) restores proteostasis and remodels specific subsets of the proteome in cells from healthy and AMD donors. CA77.1-treated AMD iPSC-RPE display reduced oxidative stress and improved mitochondrial function. These findings may explain the specific vulnerability of the RPE during AMD and shed light on CMA as a new druggable target for this as-of-now incurable disease.**

**Keywords** Chaperone-mediated Autophagy; Age-related Macular degeneration; RPE; Proteostasis; Oxidative Stress
**Subject Category** Autophagy & Cell Death

## Introduction

Age-related macular degeneration (AMD) is the leading ocular disease in the elderly population, and its progression leads to permanent central vision loss. AMD is characterized by progressive bilateral degeneration of the macula, a cone photoreceptor-rich retinal region found in the posterior part of the eye and responsible for detailed, color vision (Deng et al, 2022). Since it is inherently an age-associated disease, AMD prevalence is expected to double by 2040 due to aging populations (Wong et al, 2014), but other environmental and genetic variables such as smoking or the presence of the $CFH^{Y402H}$ (complement factor H) polymorphism have also been linked to higher AMD incidence (Landowski et al, 2019). The disease can be classified as "wet" or "dry" depending on whether it presents a neovascular component or not. Wet AMD is characterized by choroidal neovascularization that disrupts the retinal pigment epithelium (RPE) and causes local edema triggering photoreceptor cell death. Dry AMD involves the formation of a histological hallmark termed drusen, caused by progressive extracellular deposition of debris (lipids, oxidized proteins, complement, trace elements) between the RPE and Bruch's membrane. Similarly, drusen will disrupt the RPE, trigger photoreceptor cell death, and lead to what is known as geographic atrophy. Antiangiogenic immunotherapy (anti-VEGF) has been shown to effectively slow the progression of wet AMD (Deng et al, 2022). For dry AMD, treatments that target the complement pathway have shown a moderate improvement and have recently been approved for clinical use, but their long-term efficacy remains to be fully characterized (Heier et al, 2023; Khanani et al, 2023). In both AMD subtypes, the RPE undergoes primary degeneration, but the cause of this cell type-specific vulnerability remains to be elucidated.

[1]Department of Cellular and Molecular Biology, Centro de Investigaciones Biológicas Margarita Salas, CSIC, Madrid, Spain. [2]Tumour Biology Programme, Centro Nacional de Investigaciones Oncológicas (CNIO), Madrid, Spain. [3]Doheny Eye Institute, Pasadena, CA, USA. [4]Department of Biology, Faculty of Science and Medicine, University of Fribourg, Fribourg, Switzerland. [5]Department of Developmental and Molecular Biology, Albert Einstein College of Medicine, Bronx, NY, USA. [6]Institute for Aging Research, Department of Medicine, Albert Einstein College of Medicine, Bronx, NY, USA. [7]Stem Cell Institute, University of Minnesota, Minneapolis, MN, USA. [8]Department of Genetics, Cell Biology, and Development, University of Minnesota, Minneapolis, MN, USA. [9]Department of Biochemistry, Albert Einstein College of Medicine, Bronx, NY, USA. [10]Department of Medicine, Albert Einstein College of Medicine, Bronx, NY, USA. [11]Montefiore Einstein Comprehensive Cancer Center Cancer Therapeutics Program, Albert Einstein College of Medicine, Bronx, NY, USA. [12]Department of Ophthalmology, University of California, Los Angeles, Los Angeles, CA, USA. [13]Department of Neuroscience and Movement Science, Faculty of Science and Medicine, University of Fribourg, Fribourg, Switzerland. [14]Present address: IMDEA Food, Madrid, Spain. ✉E-mail: jjimenezl@cnio.es; dferrington@doheny.org; patricia.boya@unifr.ch

The study of AMD is complex since there are no animal models that fully recapitulate the disease (Soundara Pandi et al, 2021), therefore donor tissue is extremely valuable and has provided insight into the molecular mechanisms altered in AMD. Common diagnostic criteria and sample grading have helped standardize the results obtained, and the well-known classification for human donor eye-bank tissue is the AMD Minnesota Grading System (MGS) that differentiates between healthy (MGS1), early (MGS2), intermediate (MGS3), and late AMD (MGS4) (Olsen and Feng, 2004; Olsen et al, 2017). Studies using primary RPE cells derived from AMD patients point to deficient autophagy, e.g., accumulation of LC3-II (Ye et al, 2016) or abnormal lysosome morphology (Golestaneh et al, 2017), and dysfunctional mitochondria (Ferrington et al, 2017) as drivers of AMD pathogenesis in the RPE.

The term autophagy encompasses three catabolic pathways that target cargo for degradation inside the lysosome: macroautophagy, microautophagy, and chaperone-mediated autophagy (CMA) (Villarejo-Zori et al, 2021). While macroautophagy and microautophagy can also undertake the recycling of lipids, membranes, or even whole organelles, CMA is a selective pathway for protein degradation (Kaushik and Cuervo, 2018). Moreover, CMA substrates must contain a targeting KFERQ-like motif within their amino acid sequence that will be recognized by the cytosolic chaperone HSC70. Around 45% of the human proteome contains a canonical KFERQ-like motif (Kirchner et al, 2019) and post-translational modifications (PTMs), such as phosphorylation or acetylation, can also generate novel non-canonical motifs. LAMP-2A (lysosome-associated membrane protein 2, isoform A) is a result of alternative splicing of the LAMP2 gene and is the lysosomal receptor for CMA. Its isoform-specific cytosolic tail is able to dock HSC70 bound to KFERQ-containing substrates, and this event triggers LAMP-2A multimerization generating a protein translocation complex in the lysosomal membrane (Bandyopadhyay et al, 2008). After substrate protein unfolding, a form of HSC70 found in the lysosomal lumen assists in the translocation of the substrate into the lysosome where it ultimately undergoes degradation by acidic proteases (Agarrabertes et al, 1997; Cuervo et al, 1997) (Fig. 1A). Only a specific subset of LAMP-2A$^+$HSC70$^+$ lysosomes can perform CMA (Cuervo et al, 1997). Besides protein quality control, CMA contributes to modulate, through selective and timely proteome remodeling, cellular processes, such as glycolysis (Schneider et al, 2014), lipolysis (Kaushik and Cuervo, 2015), and cell cycle (Park et al, 2015), among others.

Recently, it was described that LAMP-2 is preferentially expressed in the RPE and its levels decrease with age in mice (Notomi et al, 2019). Furthermore, mice with ablation of the whole Lamp2 gene (Lamp2$^{-/-}$) present age-associated sub-RPE deposits reminiscent of the early stages of drusen formation (Notomi et al, 2019). While no studies have focused on the specific role of LAMP-2A, the only isoform required for CMA, in AMD pathogenesis, the same group has shown that the RPE of AMD donors presents decreased protein levels of total LAMP-2 compared to age-matched controls (Notomi et al, 2019).

In the present work, we show that CMA is selectively impaired in the RPE of patients with AMD leading to proteostasis failure and the accumulation of KFERQ-containing substrates. Crucially, these findings are replicated in conjunctiva-derived iPSC-RPE of donors with AMD. Proteomic studies revealed that CMA remodels shared and unique subsets of the proteome in healthy and AMD iPSC-RPE.

Pharmacological activation of CMA resolved proteostasis defects, stimulated NRF2-mediated antioxidant response, and alleviated metabolic dysfunction in iPSC-RPE from donors with AMD.

## Results

### CMA is impaired in the RPE of patients with AMD

Analysis of mRNA levels of different LAMPs in publicly available healthy human retina scRNA-seq datasets (GSE142449) (Voigt et al, 2020) revealed that, similarly to its murine counterpart (Notomi et al, 2019), LAMP2 expression in human retina is markedly higher in the RPE and glial cells (Fig. EV1 and Dataset EV1). The network of effectors, positive and negative modulators of CMA, has been extensively characterized and their expression levels can be used to infer CMA activity (CMA score) by performing a directed, weighted average (Bourdenx et al, 2021). CMA score showed a trend towards a reduction selectively in the RPE (RPE65$^+$BEST1$^+$ cells) of patients with AMD (Fig. 1B) compared to other cell types within the subretinal space (Fig. EV2A) in a publicly available RPE/choroid scRNA-seq dataset (GSE135922). Supporting a putative impairment of CMA, proteomic analyses revealed that abundance of KFERQ-containing proteins was increased in both the RPE of patients with early AMD (MGS2) as well as within drusen (Crabb et al, 2002), when compared to healthy patients (MGS1) and the reference human proteome (Fig. 1C). Furthermore, there was a significant enrichment on literature-validated CMA protein substrates (Dataset EV2) in the RPE of AMD donors (Fig. 1D and Table EV1). Finally, we also observed a significant decrease in the protein levels of both LAMP-2A and HSC70 in the RPE of histological slides from AMD patients (Figs. 1E and EV3).

CMA impairment was limited to the RPE as no major changes were observed in the neuroretina of AMD patients (Fig. EV2B; GSE135092) or in donors stratified according to MGS guidelines (Fig. EV2C; GSE115828). We have previously described that in several tissues and cell types, including the retina, CMA is modulated at the transcriptional level by the RARα transcription factor, and that the inhibitory effect of RARα on CMA can be reversed by enhancing its interaction with the co-repressor N-CoR1 (Gomez-Sintes et al, 2022). However, in the case of the neuroretina, we did not observe any changes in the protein levels of LAMP-2A or N-CoR1/RARα ratio, by immunoblotting (Fig. EV2D,E) or immunofluorescence (Fig. EV2F). While dysfunctional macroautophagy has been described in both neuroretina and RPE (Golestaneh et al, 2017; Mitter et al, 2014), herein we demonstrate for the first time selective impairment of CMA in the RPE that could be responsible for the selective vulnerability of this cell type throughout AMD progression and the cause of other cellular alterations unique to these cells such as impaired mitochondrial function (Ferrington et al, 2017) or aberrant ROS production (Datta et al, 2017).

### AMD donor-derived iPSC-RPE recapitulate CMA alterations

Human donor-derived iPSC-RPE were used as an in vitro model to further dissect the cellular and molecular mechanisms mediating

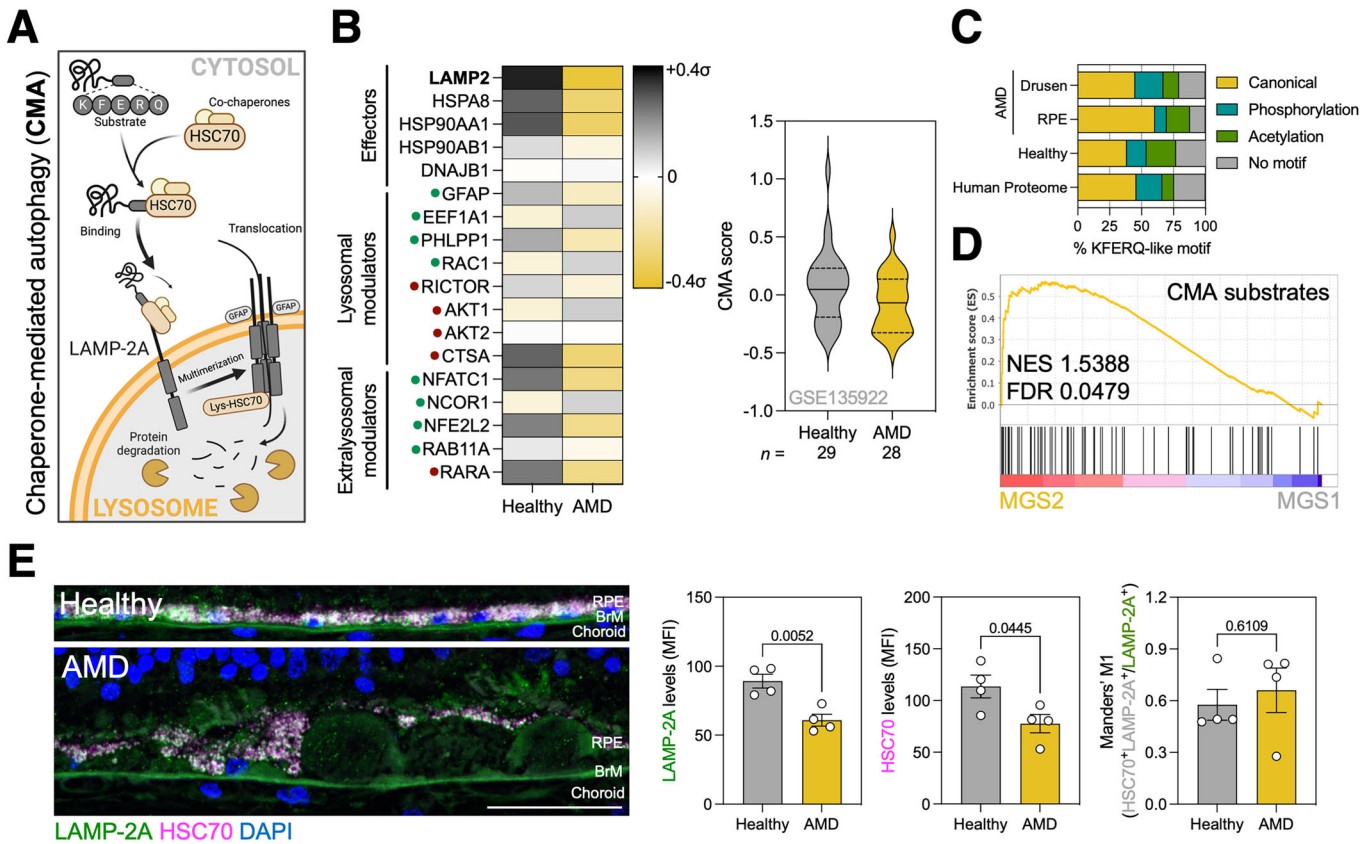

**Figure 1. CMA is impaired in the RPE of patients with AMD.**

(A) Diagram depicting the main steps involved in chaperone-mediated autophagy (CMA). (B) Heatmap showing the mRNA levels of the CMA network components (left; effectors, positive (green) and negative (red) modulators) and CMA score in the RPE cluster of a publicly available scRNA-seq dataset (GSE135922; right). (C) Abundance of canonical, acetylation- or phosphorylation-generated KFERQ-like motif-containing proteins in proteomic data from RPE of AMD patients (MGS2; PXD033413) and drusen (Crabb et al, 2002). Data are compared to the reference Human Proteome. (D) GSEA enrichment analysis of validated CMA substrates in the RPE of AMD patients (MGS2; PXD033413). (E) Representative images and quantification of donor sections immunostained against LAMP-2A (green) and HSC70 (magenta), nuclei were counterstained with DAPI (blue). Quantification of the levels of both proteins (MFI) and the proportion of lysHSC70 CMA-proficient lysosomes in the RPE is shown. ($n = 4$). Scale bar, 50 µm. All data are expressed as the mean ± s.e.m. Dots represent individual donors. $p$ values were calculated using unpaired Student's $t$ test. Source data are available online for this figure.

CMA dysregulation in AMD. Briefly, epithelial cells were obtained from the conjunctiva of MGS-classified healthy (MGS1) or AMD (MGS2-3) eyes, reprogrammed into iPSCs, and differentiated into iPSC-RPE monolayers using a standardized pipeline (Geng et al, 2017) (Fig. 2A, Table 1, and Figs. EV4A,B and Table EV2). Supporting our findings in vivo, iPSC-RPE from AMD donors presented a significantly decreased CMA score (Figs. 2B and EV4C). While no significant changes in *LAMP2A* mRNA levels were observed (Fig. EV2D), AMD donors had significantly decreased LAMP-2A protein levels (Fig. 2C,D), supporting the possibility of posttranscriptional mechanisms such as changes in LAMP-2A protein trafficking or stability contributing to the observed reduced protein levels. We also observed a significant decrease in N-CoR1/RARα ratio, mainly caused by a decrease in N-CoR1 levels (Figs. 2C,D and EV4E). To assess CMA activity in iPSC-RPE, we used the KFERQ-PS-Dendra reporter, which allows for quantitative evaluation of delivery and association of an exogenous CMA substrate to lysosomes (Fig. 2E) (Dong et al, 2020). AMD iPSC-RPE display significantly decreased levels of CMA in basal conditions (Fig. 2F). Interestingly, serum deprivation for 24 h led to an

increase in KFERQ-Dendra+ puncta in both groups and abrogated the differences between healthy and AMD iPSC-RPE (Fig. 2F). These results indicate that it is still possible to activate this pathway in acute stress conditions such as nutrient deprivation. Notably, AMD iPSC-RPE also presented a very pronounced and significant accumulation of cytosolic protein aggregates assessed by ProteoStat+ staining (Fig. 2G). We also tested the impact of different variables on key CMA readouts (CMA score, KFERQ-Dendra+ puncta, LAMP-2A levels, N-CoR1/RARα ratio; Fig. EV5). An age-associated decrease in CMA has been reported for other tissues (Cuervo and Dice, 2000; Dong et al, 2021; Kiffin et al, 2007; Madrigal-Matute et al, 2022), and we similarly observed a trend towards a decrease of all parameters across physiological aging (Fig. EV5A). No differences were observed regarding reported donor sex (Fig. EV5B). Crucially, we observed a progressive decrease in almost all CMA readouts when we sub-classified AMD iPSC-RPE as MGS2 (early AMD) or MGS3 (intermediate AMD) (Fig. EV5C). We were not able to detect a significant correlation between the high-risk CFH^Y402H variant and the analyzed CMA readouts (Fig. EV5D), but future studies including more high-risk

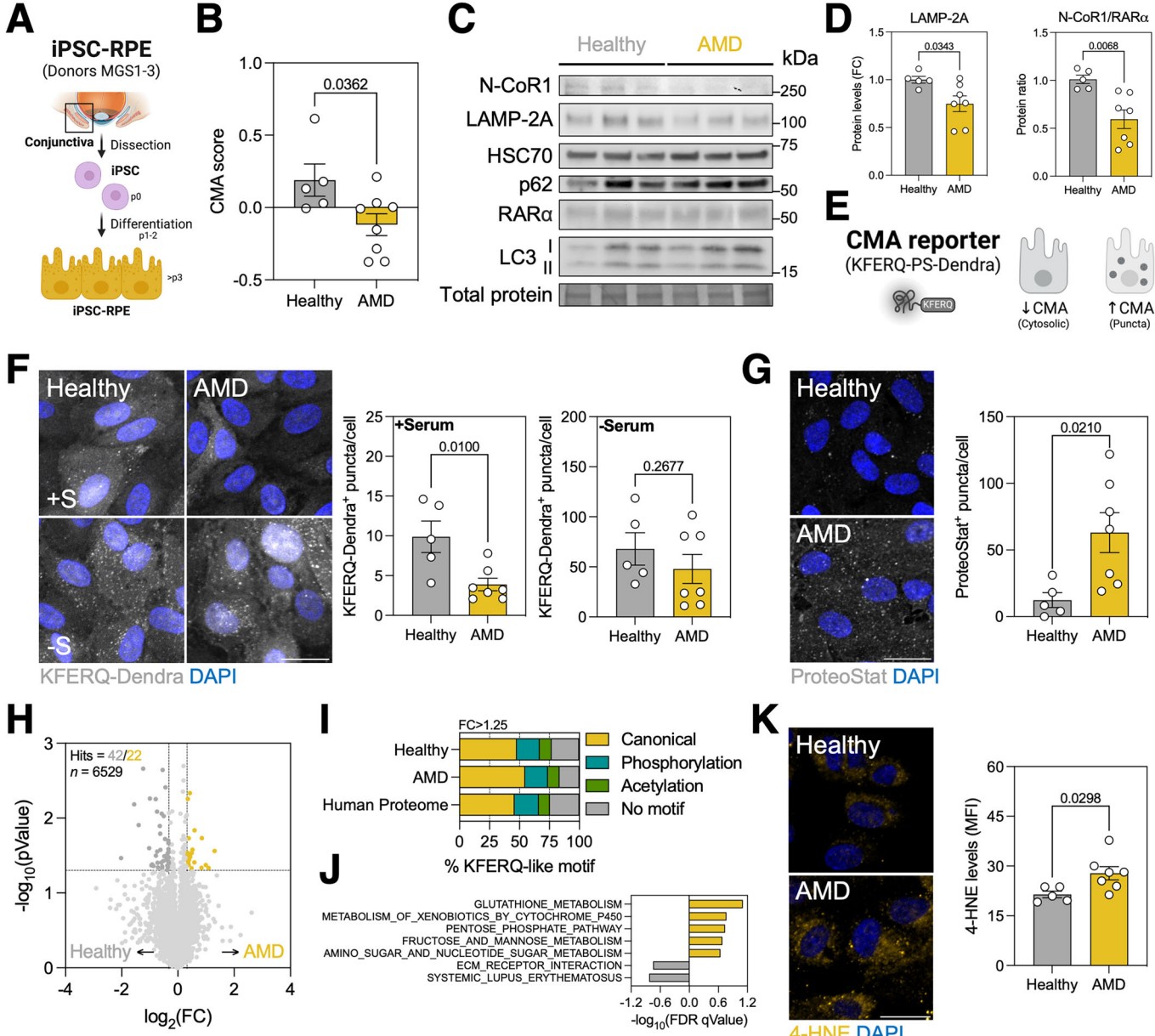

**Figure 2. AMD donor-derived iPSC-RPE recapitulate CMA impairment.**

(A) Diagram depicting the protocol used to obtain human iPSC-RPE monolayers from reprogrammed conjunctiva epithelial cells. (B) CMA score in iPSC-RPE derived from healthy or AMD donors. ($n = 5$–8). (C) Western blot analysis of CMA-related proteins (LAMP-2A, N-CoR1, RARα, HSC70) and macroautophagic substrates (p62, LC3) in iPSC-RPE. (D) Quantification of LAMP-2A levels and N-CoR1/RARα ratio as shown in (C). ($n = 5$–7). (E) Diagram showing the basis of the KFERQ-PS-Dendra reporter. Lysosomes undertaking degradation of the exogenous CMA substrate can be identified as puncta. (F) Quantification of KFERQ-Dendra+ puncta/cell in iPSC-RPE cultured in complete medium (+S) or subjected to serum starvation for 24 h (−S). ($n = 5$–7). (G) Quantification of ProteoStat+ protein aggregates in iPSC-RPE. ($n = 5$–7). (H) Volcano plot showing differentially enriched proteins in healthy (gray) or AMD (yellow) iPSC-RPE, obtained from bulk non-targeted proteomics (6529 proteins detected). (I) Abundance of canonical, acetylation- or phosphorylation-generated KFERQ-like motif-containing proteins in proteomic data from (H). Data are compared to the reference Human Proteome. (J) Significantly upregulated (yellow) and downregulated (gray) KEGG pathways in AMD iPSC-RPE based on proteomic data from (H). (K) Immunostaining analysis of 4-HNE levels (MFI; yellow) in iPSC-RPE, nuclei were counterstained with DAPI (blue). ($n = 5$–7). Scale bars, 25 μm. All data are expressed as the mean ± s.e.m. Dots represent individual donors. $p$ values were calculated using unpaired Student's $t$ test. Source data are available online for this figure.

healthy and low-risk AMD iPSC-RPE are required to make a solid conclusion.

Bulk proteomics analysis (Fig. 2H) revealed an accumulation of proteins containing canonical and phosphorylation-activated KFERQ-like motifs (Fig. 2I), reminiscent of the phenotype

observed in the RPE of early AMD patients (Fig. 1C). Gene set enrichment analysis (GSEA) using the KEGG database showed that the top enriched pathway in the proteome of AMD iPSC-RPE was glutathione metabolism, followed by carbohydrate metabolism (Fig. 2J). Supporting this finding, we also observed significantly

**Table 1. Demographics and relevant clinical information of iPSC-RPE donors.**

| Donor | Sex | Age | MGS (1–4) | ARMS2 A69S (rs10490924) | CFH Y402H (rs1061170) |
|-------|-----|-----|-----------|-------------------------|------------------------|
| Healthy1 | M | 67 | 1 | G/G | T/T |
| Healthy2 | M | 55 | 1 | G/G | T/T |
| Healthy3 | M | 54 | 1 | G/G | T/T |
| Healthy4 | F | 77 | 1 | G/G | T/T |
| Healthy5 | F | 62 | 1 | G/G | C/C |
| AMD1 | M | 83 | 3 | G/G | C/C |
| AMD2 | F | 83 | 3 | G/G | C/C |
| AMD3 | F | 72 | 3 | G/G | C/C |
| AMD4 | F | 75 | 3 | G/G | C/C |
| AMD5 | M | 75 | 2 | G/G | T/T |
| AMD6 | F | 67 | 2 | G/G | C/C |
| AMD7 | F | 83 | 2 | G/T | T/T |
| AMD8 | M | 84 | 3 | T/T | C/C |
| AMD9 | F | 80 | 2 | G/T | T/T |

*M* male, *F* female, *MGS* Minnesota Grading System. For more information, please refer to Table EV2.

increased levels of lipid peroxidation in the AMD group, evaluated using 4-hydroxynonenal (4-HNE) immunofluorescence (Fig. 2K). To identify which of the observed increases in protein abundance in AMD were due to reduced degradation in lysosomes, we performed proteomics of iPSC-RPE treated with a combination of lysosomal proteolysis inhibitors (20 mM $NH_4Cl$ and 100 μM Leupeptin; N/L). There were 542 and 581 differentially enriched proteins (DEP) in the healthy and AMD groups, respectively (Fig. 3A). We further filtered these hits based on the fold-change (FC > 1.25), $p$ value ($p < 0.01$) and the presence of a KFERQ-like motif, narrowing down the number of hits to 171 (Healthy) and 328 (AMD) (Fig. 3B). Interestingly, there were both unique and shared putative CMA substrates for each of the groups (Fig. 3B; Jaccard index = 0.3097), indicating that CMA may regulate different functions in each group (Fig. 3B). Over-representation analysis (ORA) showed that both healthy and AMD iPSC-RPE may employ lysosomal degradation to modulate cholesterol metabolism and nutrient sensing (Fig. 3C). Healthy iPSC-RPE showed preferential lysosomal degradation of proteins involved in glycolysis and metabolic adaptation, while AMD iPSC-RPE showed lysosomal degradation of proteins involved in fatty acid metabolism, ferroptosis, and ephrin signaling (Fig. 3C).

In order to validate these findings and investigate their degradation by CMA, we compared the levels of representative proteins in the presence of a specific macroautophagy inhibitor (10 μM MRT68921) or total lysosomal proteolysis inhibition (N/L) as previously described (Juste and Cuervo, 2019) (Figs. 3D,E and EV4F). HK2 (Hexokinase 2) initiates glycolysis by phosphorylating glucose to generate glucose-6-phosphate, it has previously been shown that it can undergo CMA (Xia et al, 2015) and this was also the case in healthy iPSC-RPE (Fig. 3E,F). Surprisingly, we did not observe N/L-induced accumulation of GAPDH (glyceraldehyde 3-phosphate dehydrogenase), traditionally used to monitor CMA,

highlighting the cell-type specificity of CMA substrates (Kaushik and Cuervo, 2018). As previously mentioned, the healthy and AMD donor-derived cells shared a pool of CMA substrates with a high representation of cholesterol metabolism (Fig. 3C). We observed similar levels of CMA of SQLE (Squalene monooxygenase), that catalyzes cholesterol biosynthesis, in both groups, although the fraction of lysosomal degradation insensitive to MRT inhibition was markedly lower in AMD, supporting reduced CMA in this group (Fig. 3E,F). Interestingly, using the recently developed KFERQ-Finder tool (Kirchner et al, 2019) we identified two putative acetylation-generated KFERQ-like motifs in the sequence of SQLE (Fig. 3E). Considering the high levels of lipid peroxidation (Fig. 2K) and representation of ferroptosis-related proteins observed in ORA analysis (Fig. 3C), we assessed CMA of ACSL4 (Acyl-CoA Synthetase Long chain family member 4). Lipid remodeling mediated by ACSL4 can propel ferroptosis execution (Doll et al, 2017). It has been previously validated as a CMA substrate (Liu et al, 2022), and in accordance, we found that cellular levels of ACSL4 were markedly elevated and lysosomal degradation of ACSL4 was stalled in iPSC-RPE from donors with AMD (Fig. 3E,F).

In addition, we interrogated whether macroautophagy and the ubiquitin-proteasome system were being upregulated in response to CMA deficiency in AMD iPSC-RPE, as previously reported in other cell types (Massey et al, 2006). First, we measured macroautophagy flux ($LC3-II_{N/L}/LC3-II_{Control}$) and observed no differences between healthy and AMD iPSC-RPE (Figs. 3E and EV6A). No alterations were observed neither in lysosomal function, evaluated using fluorogenic Cathepsin B substrate (Fig. EV6B), nor in acidic lysosomal mass (Fig. EV6C). To obtain subcellular resolution and detect more subtle alterations, we performed immunofluorescence staining of autophagic cargo (p62, magenta), autophagic vacuoles (LC3, cyan), or endolysosomes (LAMP-1, yellow) and analyzed single-, double-, or triple-positive vesicles (Fig. EV6D). No differences were detected in any of the parameters analyzed (Fig. EV6E–K). In addition, we also performed a recently developed targeted proteomics approach that offers robust quantification of autophagy-related proteins, including several selective autophagy receptors (SAR) (Leytens et al, 2025). Healthy and AMD iPSC-RPE showed similar flux of ATG8-family proteins (GABARAPL1), mitophagy receptors (BNIP3) and selective autophagy adaptors (CALCOCO2/NDP52, SQSTM1/p62, TAX1BP1) upon lysosomal degradation inhibition with N/L (Fig. EV6L), indicating that macroautophagy is indeed fully functional in both backgrounds. Finally, we also measured the levels of K48-linked poly-ubiquitinated proteins, the canonical targeting signal for degradation via ubiquitin–proteasome system, but observed no differences between healthy and AMD iPSC-RPE (Fig. EV6M).

Our findings suggest a selective impairment of CMA in iPSC-RPE derived from donors with AMD, phenocopying our initial observations in AMD donor tissue (Fig. 1). Accumulation of specific proteins or rerouting of others to alternative protein disposal pathways also play a role in RPE dysfunction during AMD etiopathogenesis. Notably, no compensatory upregulation or crucial alterations were observed in macroautophagy or ubiquitin-proteasome, highlighting CMA deficiency as the main driver of the observed changes in lysosomal protein degradation and of the subsequent proteotoxicity in AMD iPSC-RPE.

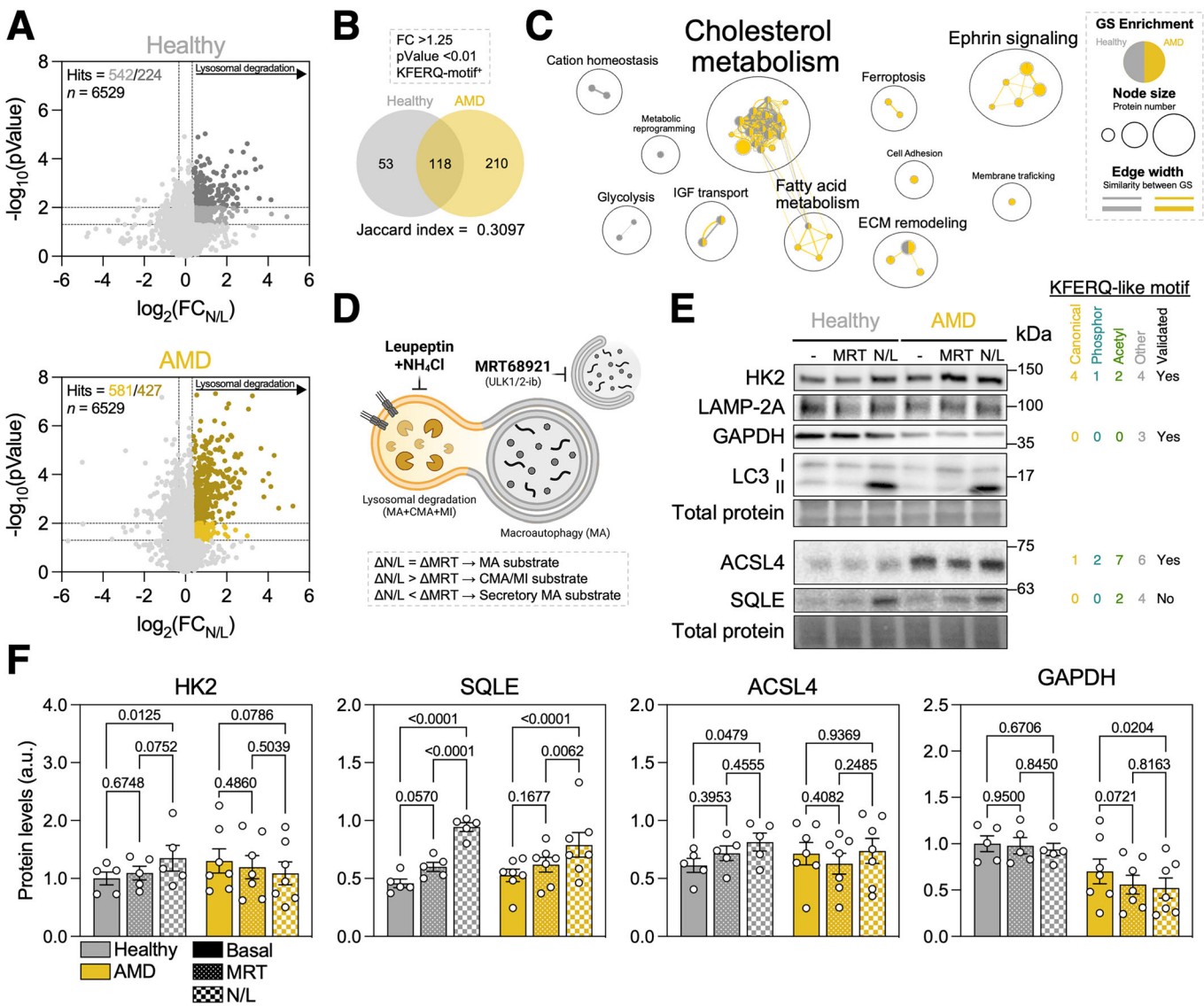

**Figure 3. CMA regulates different subsets of the proteome in healthy and AMD iPSC-RPE.**

(A) Volcano plot showing differentially enriched proteins in healthy (light ($p < 0.05$) and dark ($p < 0.01$) gray; top) or AMD (light ($p < 0.05$) and dark ($p < 0.01$) yellow; bottom) iPSC-RPE treated for 24 h with 20 mM NH4Cl and 100 μM Leupeptin (N/L) to inhibit lysosomal proteolysis. (B) Venn diagram of putative CMA substrates (defined as fold change >1.25, $p$ value < 0.01, and presence of at least one KFERQ-like motif) enriched in healthy and AMD iPSC-RPE treated with N/L. Jaccard index represents the overlap between both conditions. (C) Annotated network visualization obtained from over-representation analysis (ORA; KEGG, WikiPathways, Reactome databases) of putative CMA substrates in N/L-treated iPSC-RPE. Node and edge legend is shown. (D) Diagram depicting the experimental design used to assess the degradation of specific substrates by different autophagy pathways. Treatment with 10 μM MRT68921 inhibits autophagosome formation, and therefore macroautophagy (MA). Treatment with N/L inhibits all lysosomal proteolysis, including MA, CMA, and microautophagy (MI). Medium was replaced the day before the experiment, and iPSC-RPE were incubated in the presence of the inhibitors for 6 h. (E) Western blot analysis of putative CMA substrates (HK2, GAPDH, ACSL4, SQLE) and autophagy proteins (LAMP-2A, LC3-I/II) in iPSC-RPE. Presence of KFERQ-like motif and evidence of CMA-mediated degradation in the literature is shown (right). (F) Quantification of protein levels as shown in (E). ($n = 5$–7). All data are expressed as the mean ± s.e.m. Dots represent individual donors. $p$ values were calculated using unpaired Student's $t$ test (A) or two-way RM ANOVA with Tukey's post hoc test (F). Source data are available online for this figure.

## Pharmacological activation of CMA restores proteostasis and improves cellular fitness

RARα-mediated retinoic acid (RA) signaling has been previously shown to negatively modulate CMA by activating the transcription of negative CMA regulators (Anguiano et al, 2013; Gomez-Sintes et al, 2022). Our group recently developed a new generation of CMA-activating compounds that inhibit RARα by stabilizing its

interaction with co-repressor N-CoR1 (Fig. EV7A) (Gomez-Sintes et al, 2022). Treatment with the lead compound CA77.1 (10 μM) increased the CMA score in 4/5 healthy iPSC-RPE and all AMD iPSC-RPE cell lines analyzed (Fig. EV7B,C). Most importantly, CA77.1 treatment increased CMA activity (KFERQ-Dendra+ puncta) (Fig. 4A) and resolved proteostasis defect in AMD iPSC-RPE (Fig. 4B). As expected, treatment with CA77.1 significantly decreased cellular levels of a fraction of the proteome (105 proteins)

in the healthy iPSC-RPE, and this number doubled in the AMD iPSC-RPE (204 proteins displayed CA77.1-induced decrease in cellular levels) in further support of a protein backlog in AMD cells before treatment (Fig. 4C). Interestingly, we also found a fraction of proteins whose cellular levels increase upon CA77.1 treatment. We reasoned that these cannot be changes due to increased direct degradation by CMA but could be a consequence of the removal of other proteins and subsequent decrease in proteotoxicity (Fig. 4B). GSEA revealed a very robust upregulation of mitochondrial and peroxisomal biogenesis (Figs. 4D and EV8). RPE relies heavily on fatty acid-fueled mitochondrial metabolism due to the daily digestion of up to 10% of photoreceptor outer segments (Jiménez-Loygorri et al, 2023), and a decrease in mitochondrial respiration has been previously described in AMD patients (Ferrington et al, 2017). To validate this finding, we measured bulk mitochondrial mass using MitoTracker Green (MTG) live imaging and observed a significant increase in CA77.1-treated iPSC-RPE (Fig. 4E,F). Similarly, we also observed increased levels of structural and oxidative phosphorylation-related mitochondrial proteins in our proteomics study (Fig. EV8). Microplate respirometry using Seahorse technology showed that AMD cells treated with CA77.1 displayed increased mitochondrial maximal respiration, reaching the levels of healthy iPSC-RPE (Fig. 4G). Because we observed that some glycolytic enzymes undergoing CMA degradation in healthy iPSC-RPE were no longer degraded in the AMD group (Fig. 3), we analyzed the effect of the CMA activator in glycolysis. Pharmacological activation of CMA with CA77.1 mildly, but significantly, decreased glycolytic activity in healthy iPSC-RPE but had no effect in the AMD group (Fig. 4H), indicating that restoration of proteostasis through boosting CMA differentially modulates glycolysis- and mitochondria-driven ATP production.

AMD iPSC-RPE presented a very robust increase in 4-HNE levels (Fig. 2K), and treatment with CA77.1 induced a significant increase in the mRNA levels of the transcription factor *NFE2L2*/NRF2 (Figs. 5A and EV7B), a master regulator of the antioxidant response and positive modulator of CMA (Pajares et al, 2018a). NRF2 immunostaining revealed decreased nuclear translocation in AMD iPSC-RPE, that was corrected upon exposure to CA77.1 (Fig. 5B). Consequently, we also observed increased mRNA levels of downstream target genes *NQO1* and *UCP2* (Fig. 5C). Activation of NRF2-mediated antioxidant response translated into decreased levels of 4-HNE$^+$ lipid peroxidation (Fig. 5D), which might also be enhanced by restored CMA-mediated degradation of ACSL4 (Figs. 3F and 5E). Both sustained ROS production due to defective NRF2 activation and defective CMA have previously been linked to double-strand breaks and DNA damage (Gruosso et al, 2016; Park et al, 2015), we detected significantly high levels of DNA damage (~25% γH2AX$^+$ cells) in AMD iPSC-RPE that were decreased to the levels of healthy iPSC-RPE in the presence of CA77.1 (Fig. 5F).

In conclusion, pharmacological activation of CMA with CA77.1 restores proteostasis, improves mitochondrial function, and decreases oxidative stress in iPSC-RPE derived from donors with AMD.

## Discussion

Although macroautophagy has been extensively investigated in the context of AMD (Villarejo-Zori et al, 2021), to our knowledge, no

study had previously addressed the contribution of CMA to RPE homeostasis and during disease progression despite reports of reduced levels of LAMP-2. Our results show that CMA is impaired in the RPE of AMD patients, leading to the accumulation of KFERQ-like motif-containing proteins. Despite originating from conjunctiva-derived epithelial cells and undergoing reprogramming, AMD donor-derived iPSC-RPE recapitulate CMA impairment and present chronic proteotoxicity and oxidative damage. Proteomic analyses reveal that healthy and AMD iPSC-RPE resort to CMA for the degradation of proteins involved in different metabolic functions and signaling pathways. Most importantly, treatment with the pharmacological CMA activator CA77.1 increases *LAMP2* expression, activates CMA, restores proteostasis in AMD iPSC-RPE, promotes cellular energetic homeostasis, and shall be further explored as a therapeutic target for the treatment of AMD.

All-encompassing *LAMP2* deficiency (leading to lack of all splicing variants, LAMP-2A/B/C) had previously been studied in the context of Danon disease, a complex dominant genetic disorder caused by *LAMP2* mutations and characterized by severe cardiomyopathy (Endo et al, 2015). Even though less understood, clinicians have described signs of retinopathy in Danon disease mutation carriers (Kousal et al, 2021) and affected patients (Schorderet et al, 2007), some of them being indicative of primary RPE degeneration preluding photoreceptor cell death (O'Neil et al, 2022). Retinal degeneration in *Lamp2*$^{-/-}$ mice is mainly attributed to the rapid degeneration of the RPE, whereby they show signs of altered proteostasis and progressive accumulation of extracellular material in the subretinal space (Notomi et al, 2019). Previous research had shown decreased levels of LAMP-2 in the RPE of AMD patients (Notomi et al, 2019), which we now have shown to be, at least partially, due to a decrease in the alternative splicing product and limiting CMA effector LAMP-2A. Decreased LAMP-2A protein levels were also observed in AMD donor-derived iPSC-RPE, as well as decreased basal levels of CMA, but we were able to activate this pathway using either physiological (nutrient deprivation) or pharmacological (CA77.1) stimuli. Under basal conditions, AMD donor-derived cells might rely on other fully functional proteolytic pathways (macroautophagy, ubiquitin–proteasome system) to sustain homeostasis in control conditions.

Previous work has shown that CMA can serve as a compensatory mechanism in the neuroretina under circumstances where macroautophagy is impaired (Villarejo-Zori et al, 2021), such as physiological aging (Rodriguez-Muela et al, 2013), *retinitis pigmentosa* (RP) (Gomez-Sintes et al, 2022), or diabetic retinopathy (Liu et al, 2022). Evidence for the role and regulation of CMA in RPE physiology and pathology is scarcer. *LAMP2* knockdown in RPE-derived ARPE-19 cell line and human fetal RPE (hfRPE), which eliminates all LAMP-2 protein isoforms, has been shown to sensitize cells to oxidative stress, enhance lipid peroxidation, and promote ROS-induced ferroptosis (Lee et al, 2020). Furthermore, the authors also found alterations in glutathione metabolism and GSH depletion but no changes in macroautophagic flux (Lee et al, 2020). While it cannot be ruled out that these alterations are due to non-CMA-related functions of LAMP-2B/C, the phenotype of *LAMP2*-KD cells resembles our observations in AMD iPSC-RPE.

Previous evidence in the literature had shown altered macroautophagy in the primary human adult RPE (haRPE) model of donors with AMD (Golestaneh et al, 2017), but this was not the

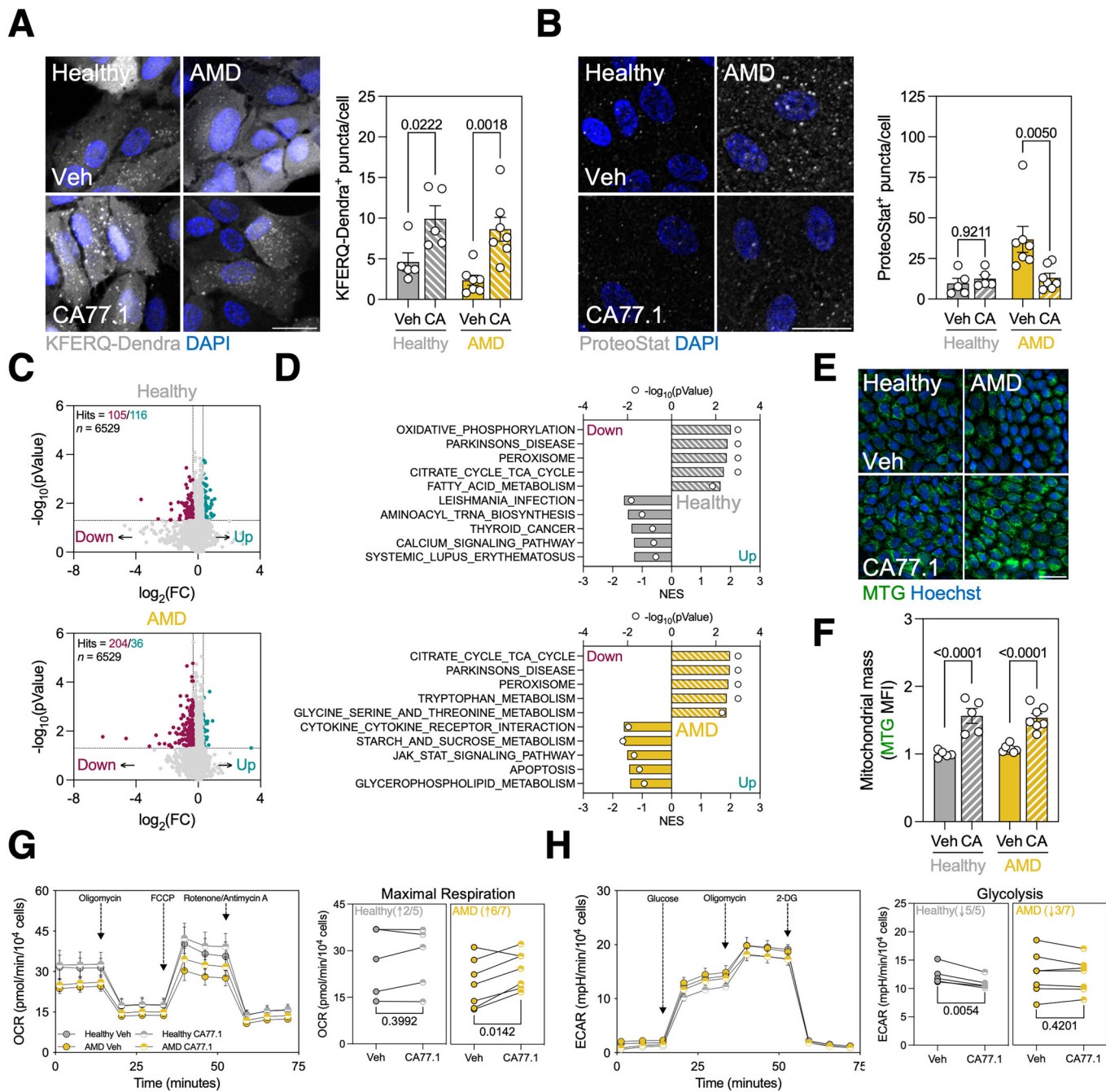

**Figure 4. Pharmacological activation of CMA restores proteostasis and improves metabolic fitness.**

(A) Quantification of KFERQ-Dendra+ puncta/cell in iPSC-RPE treated with 10 µM CA77.1 for 24 h. ($n = 5-7$). (B) Quantification of ProteoStat+ protein aggregates in iPSC-RPE treated with CA77.1. ($n = 5-7$). (C) Volcano plot showing differentially enriched (blue) or decreased (red) proteins iPSC-RPE treated with CA77.1, data obtained from bulk non-targeted proteomics (6529 proteins detected). (D) Top upregulated and downregulated KEGG pathways in CA77.1-treated healthy (top) and AMD (bottom) iPSC-RPE cells. (E, F) Representative images (E) and quantification (F) of mitochondrial mass using MitoTracker Green, analyzed by live imaging. ($n = 5-7$). (G) Mitochondrial respirometry analysis in iPSC-RPE treated with CA77.1 using Seahorse XFe96 after sequential injection of Oligomycin, FCCP, and Rotenone + Antimycin. Oxygen consumption rate (OCR) was normalized to cell number and graphs show Maximal respiration after FCCP injection. ($n = 5-7$). (H) Extracellular acidification analysis in iPSC-RPE treated with CA77.1 using Seahorse XFe96 after sequential injection of Glucose, Oligomycin, and 2-Deoxyglucose. Extracellular acidification rate (ECAR) was normalized to cell number and graphs show Basal glycolysis. ($n = 5-7$). Scale bar, 25 µm. All data are expressed as the mean ± s.e.m. Dots represent individual donors. p values were calculated using RM two-way ANOVA with Šídák's post hoc test (A, B, E) or paired Student's t test (G, H). Source data are available online for this figure.

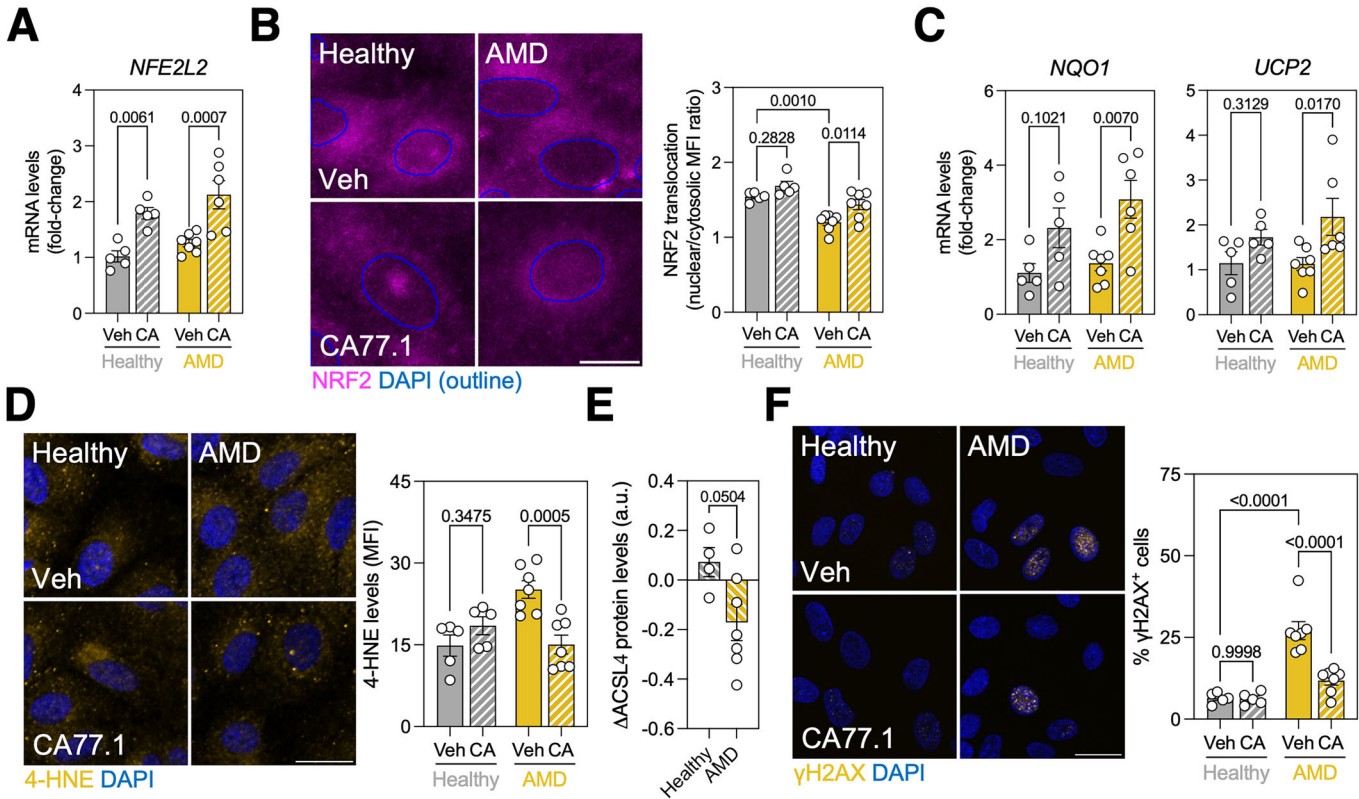

**Figure 5. CMA activation stimulates NRF2-mediated antioxidant response and alleviates oxidative damage.**

(A) Quantification of mRNA levels of *NFE2L2* in iPSC-RPE treated with 10 μM CA77.1 for 24 h. (n = 5–7). (B) Immunostaining of transcription factor NRF2 (magenta) in iPSC-RPE treated with CA77.1, nuclei were counterstained with DAPI (blue). Translocation levels are reported as the ratio between nuclear and cytosolic levels (MFI). (n = 5–7). (C) Quantification of mRNA levels of downstream targets of NRF2 activation (*NQO1*, *UCP2*) in iPSC-RPE treated with CA77.1. (n = 5–7). (D) Immunostaining analysis of 4-HNE levels (MFI; yellow) in iPSC-RPE treated with CA77.1, nuclei were counterstained with DAPI (blue). (n = 5–7). (E) Quantification of ACSL4 degradation in iPSC-RPE treated with CA77.1 (ΔACSL4 = ACSL4$_{CA77.1}$ − ACSL4$_{Basal}$). (n = 4–7). (F) Immunostaining analysis of DNA double-strand breaks in iPSC-RPE treated with CA77.1 reported as the % γH2AX+ cells (yellow), nuclei were counterstained with DAPI (blue). (n = 5–7). Scale bar, 25 μm. All data are expressed as the mean ± s.e.m. Dots represent individual donors. *p* values were calculated using two-way ANOVA with Šídák's post hoc test (A–D, F) or unpaired Student's *t* test (E). Source data are available online for this figure.

case in AMD iPSC-RPE. These differences may be due to the lysosomal alterations found in haRPE (Golestaneh et al, 2017), which undertake the daily degradation of lipid-rich photoreceptor outer segments (Villarejo-Zori et al, 2021). Our iPSC-RPE model therefore allows us to differentiate between the phenotypic alterations triggered by faulty outer segment recycling and those caused by the genetic makeup of different AMD donors.

AMD donor-derived iPSC-RPE cells indeed present functional macroautophagy but decreased levels of the CMA transcriptional network, including the receptor LAMP-2A, concomitant with impaired antioxidant response, increased lipid peroxidation, and accumulation of ferroptosis mediators known to undergo degradation via CMA, such as ACSL4. The observed decrease in CMA activity due to reduced LAMP-2A levels may result in rerouting of some of the substrate proteins to endosomal microautophagy (eMI), a pathway that similarly degrades KFERQ motif-containing proteins bound to HSC70, but in this case in late endosomes (Krause et al, 2023). Uncoordinated and untimely protein accumulation/degradation may lead to functional deregulation and hinder the capacity of the RPE to cope with physiological

(photoreceptor outer segment recycling) or environmental (hypertension, smoking) stressors (Fisher and Ferrington, 2018).

Treatment with brain-permeable CA77.1 has shown promising results in the treatment of neurodegenerative diseases associated with impaired proteostasis in experimental mouse models. Daily oral CA77.1 administration (30 mg/kg) significantly reduced protein aggregation and improved cognitive function in mouse models of tauopathy (PS19) and Alzheimer's disease (3×Tg-TauPS2APP) (Bourdenx et al, 2021). As previously mentioned, AMD research has been challenging due to the lack of animal models that recapitulate the slow progression of the disease. Mice deficient for proteins involved in mitochondrial homeostasis and antioxidant response have been shown to recapitulate some hallmarks of AMD but fail to induce drusen formation and RPE cell death (Felszeghy et al, 2019; Datta et al, 2023). Similarly, mice carrying the high-risk CFH$^{Y402H}$ allele only develop basal laminar deposits and present slightly reduced visual function (Datta et al, 2023). Therefore, there is a need to develop appropriate pre-clinical models of dry AMD as the one utilized in this study. Nonetheless, and in support of the potential suitability of our intervention

in vivo, we have demonstrated in previous studies that activation of CMA by stabilizing N-CoR1/RARα interaction has a neuroprotective effect in the *rd10* mouse model of RP (Gomez-Sintes et al, 2022). Macroautophagy is also impaired in *rd10* mice due to lysosomal alterations like those observed in AMD patients (Mitter et al, 2014) and primary RPE cultures (Golestaneh et al, 2017). Remarkably, a single intravitreal (IVT) injection with CA77.1 (40 µM/1 µL) increased photoreceptor cell survival and alleviated visual function loss in *rd10* mice (Gomez-Sintes et al, 2022). Since IVT injection is currently the *gold standard* drug administration route for AMD and other progressive ocular diseases in the clinic (Deng et al, 2022), IVT (or oral) administration of CA77.1 warrants further exploration in complementary studies in the context of AMD management.

We found it interesting that CA77.1 administration to AMD iPSC-RPE cells led to transcriptional upregulation of NRF2. Although future studies will be required to determine if this effect could be in part a direct result of the drug, we favor the idea that restoration of CMA upon CMA77.1 administration could promote degradation of repressors of NRF2 expression, such as Keap1, recently identified as a bona fide CMA substrate (Zhu et al, 2022). Since we have previously found that NRF2 upregulates *Lamp2a* expression (Pajares et al, 2018b), we propose that the axis NRF2/LAMP-2A could serve as a feedforward positive loop, thus sustaining CMA activation.

In summary, our work highlights an RPE-specific downregulation in CMA that may account, at least partially, for the proteostatic alterations observed during AMD progression. This phenomenon is conserved in iPSC-RPE derived from donors with AMD and is susceptible to pharmacological modulation using the CMA activator CA77.1, which alleviates several hallmarks of RPE degeneration.

# Methods

### Human samples

Donor data for publicly available transcriptomic and proteomics data are provided in Dataset EV1. Formalin-fixed and paraffin-embedded (FFPE) macular sections from anonymous donors were obtained from Lion's Gift of Sight Eye Bank (Minnesota, USA). De-identified samples were obtained with informed consent of the donor or donor's family and in accordance with the WMA's Declaration of Helsinki and the Department of Health and Human Services Belmont Report. The Lion's Gift of Sight Eye Bank is licensed by the Eye Bank Association of America (Accreditation #0015204) and accredited by the FDA (FDA Established Identifier 3000718538). De-identified donor tissue is exempt from the process of Institutional Review Board approval. Donors were classified as Healthy or AMD by a trained ophthalmologist, and additional clinical observations are included in Table EV1.

### Donor-derived iPSC-RPE obtention

Conjunctiva from de-identified donors was similarly obtained from Lion's Gift of Sight Eye Bank (Minnesota, USA). Donors were classified as Healthy (MGS1) or AMD (MGS2-3) by a Board-Certified Ophthalmologist, and additional clinical observations, including the Minnesota Grading System (MGS) score, are included in Table EV2. Reprogramming of conjunctival epithelial cells into hiPSCs was performed using the CytoTune™ 2.0 Sendai Reprogramming Kit (A16517, Thermo Fisher) following the manufacturer's instructions as previously described (Geng et al, 2017).

### Donor-derived iPSC-RPE differentiation

Cell culture plates were coated with Matrigel (356234, Corning). From P0-P2, cells were maintained in TheraPEAK X-VIVO-10 serum-free cell culture medium (BP04-743Q, Lonza) containing 1× Pen/strep antibiotics (15070-063, Gibco) and supplemented with 10 µM Y-27632 ROCK inhibitor (S1049, SelleckChem) for the first week or with 10 mM Nicotinamide (72340, Merck) for the rest. For >P2, cells were maintained in MEMα+GlutaMAX (32561, Gibco) containing 1× Pen/Strep, 1× N1 (N6530, Merck), 1× non-essential amino acids (11140050, Gibco), 250 mg/L taurine (T0625, Merck), 55 nM hydrocortisone (H6909, Merck) and 6.5 ng/L triiodo-L-thyronine (T5516, Merck) supplemented with 5% FBS (S11150H, R&D Systems) and 10 µM Y-27632 for the first week or with 2% FBS and 10 mM Nicotinamide for the remaining time in culture. Cells were passaged every 4 weeks using Accumax (00-4666-56, Thermo Fisher), and the medium was changed twice a week. All experiments were performed at P3-P4 and at least 2 weeks after the last passage. For assays analyzing mitochondrial function or metabolism, cells were cultured without nicotinamide.

To physiologically induce autophagy, cells were subjected to starvation in MEMα+GlutaMAX without any supplements or FBS for 24 h. To analyze the contribution of different pathways to substrate protein degradation, cells were treated with 10 µM MRT68921 (S7949, SelleckChem) or 100 µM Leupeptin (L2884, Merck) and 20 mM NH$_4$Cl (A9434, Merck) for 6 h. To pharmacologically induce CMA, cells were treated with 10 µM CA77.1 (SML3197, Merck) for 24 h as previously reported (Gomez-Sintes et al, 2022).

Cells were routinely tested for mycoplasma contamination, and iPSC-RPE cell identity was confirmed based on cobblestone-like morphology and expression of prototypic RPE proteins (Fig. EV4A,B).

### Immunohistochemistry and immunocytochemistry

FFPE slides (4 µm thickness) were generated by a trained technician at Lion's Gift of Sight Eye Bank (Minnesota, USA). Hematoxylin/eosin staining was performed, and histopathological observations and diagnosis were carried out by a certified ophtalmologist (Table EV1). FFPE slides were deparaffinized and rehydrated using standard protocols (3×Xylene, 100% EtOH, 96% EtOH, 70% EtOH, dH$_2$O; 5 min each), then subjected to Tris-based antigen retrieval performed according to manufacturer's instructions (H-3301-250, Vector Labs). Samples were permeabilized with 0.3% Triton X-100 (BP151-100, Fisher Scientific) for 20 min, then blocked with NGS buffer (10% NGS (G9032, Merck), 0.1% Triton X-100 in PBS) for one hour at RT. Primary antibodies (Table EV3) were diluted in NGS buffer and incubated overnight at 4 °C. After three 5-min washes with PBS, secondary antibodies (Table EV3) were similarly diluted in NGS buffer and incubated for 1 h at RT. To counterstain

nuclei, 1 µg/mL DAPI (D9542, Merck) was added and incubated together with the secondary antibodies. After three 5-min washes with PBS, samples were mounted using Vectashield (H-1000-10, Vector Labs). Confocal imaging was performed using a Zeiss LSM 510 (Zeiss) confocal microscope equipped with a ×63 immersion objective. Negative (secondary antibody-only) and positive controls (ARPE-19 cells) are included in Fig. EV2.

For immunofluorescence staining of iPSC-RPE, $2 \times 10^4$ cells per well were seeded over Matrigel-coated cell culture slides. Cells were carefully washed twice with PBS and fixed using 4% paraformaldehyde for 15 min at RT. Samples were permeabilized/blocked using BGT buffer (BSA-Glycine-Triton; 3 mg/mL BSA, 0.25% Triton X-100, and 100 mM glycine (1610718, Bio-Rad) in PBS) for 20 min. Primary antibodies (Table EV3) were diluted in BGT buffer and incubated for 1 h at RT. After three 5-min washes with PBS, secondary antibodies (Table EV3) were similarly diluted in BGT buffer and incubated for 1 h at RT. To counterstain nuclei, 1 µg/mL DAPI was added and incubated together with the secondary antibodies. To measure protein aggregate accumulation, the ProteoStat assay (ENZ-51035-K100, Enzo) was performed following the manufacturer's instructions. After three 5-min washes with PBS, samples were mounted using ProLong Diamond (P36970, Merck) and cured overnight. Confocal imaging was performed using a Zeiss LSM 510 (Zeiss) confocal microscope equipped with a ×63 immersion objective.

## RT-qPCR

In all experiments, $2.5 \times 10^5$ cells per well were seeded over Matrigel-coated 24-well standard culture plates. Cells were washed twice with sterile PBS and total RNA was isolated using QIAGEN RNeasy Mini kit (74104, QIAGEN) according to the manufacturer's instructions. To obtain cDNA, 1 µg of RNA was retrotranscribed using SuperScript III First-Strand Synthesis System (18080051, Thermo Fisher). Transcript mRNA levels were determined by RT-qPCR using either PowerUp SYBR Green (A25780, Thermo Fisher) with in-house primers (Table EV4) or TaqMan probes (Table EV5) with NZY Speedy qPCR Probe Master Mix (MB23003, NZYtech). Ribosomal *18S* was included as a housekeeping gene in all experiments.

## CMA transcriptional score

Using either RT-qPCR or publicly available RNA-seq data, the CMA score was calculated by averaging the weighted and directed transcript levels of known CMA effectors and positive and negative regulators as previously described (Bourdenx et al, 2021). Briefly, counts were log2-normalized, *LAMP2*, since it is the limiting effector in the pathway, was given a weight of 2, and the rest of the components −1 or 1 based on their effect on CMA activity.

## Immunoblotting

In all experiments, $5 \times 10^5$ cells per well were seeded over Matrigel-coated 12-well standard culture plates. Total protein was collected using ice-cold RIPA lysis buffer (R0278-50, Merck) supplemented with protease (P8340, Merck) and phosphatase (P5726, Merck) inhibitors. Protein concentration was measured using Pierce BCA

Protein assay (A55865, Thermo Fisher). Samples were diluted, supplemented with 4× Laemmli Sample buffer (1610747, Bio-Rad) and briefly boiled before loading them on precast 4–15% Mini-PROTEAN® TGX (4561086, Bio-Rad) or AnykD Criterion TGX Midi gels (5671125, Bio-Rad). Protein was transferred onto 0.2 µm PVDF membranes (1704272, Bio-Rad) using a TransBlot semi-dry transfer system (Bio-Rad). Transfer efficiency was assessed using Ponceau S staining (78376, Merck). Membranes were blocked with 5% BSA (A30075-100.0, RPI) or milk in TBS-T (0.1% Tween-20 (1706531, Bio-Rad) in PBS) for 1 h and incubated overnight at 4 °C with primary antibodies diluted 1:1000 (Table EV3) in 5% BSA in PBS. Secondary antibodies were diluted 1:2000 in TBS-T (Table EV3) and incubated for 1 h at RT. Membranes were developed using Pierce ECL Western Blotting substrate (32106, Thermo Fisher) or SuperSignal West Dura (34076, Merck) and x-ray film (AGFA) using a CURIX 60 Processor (AGFA). High-abundance proteins were detected using a Chemidoc imaging system (Bio-Rad). After detecting all the pertinent antibodies, total protein within the membranes was stained again using Bio-Safe Coomassie Stain (1610786, Bio-Rad) and air-dried.

## CMA activity analysis using KFERQ-PS-Dendra

HEK293T cells (CRL-3216; ATCC) were transfected with VSV-G, pMDLg/pRRE, REV, and KFERQ-PS-Dendra plasmids using CaCl$_2$ method. After 24 h, supernatant was collected, centrifuged at $2000 \times g$ for 3 min, and filtered using a 0.45 µm syringe filter. For transient lentiviral transduction, $2 \times 10^4$ iPSC-RPE cells per well were seeded over Matrigel-coated cell culture slides (81816, Ibidi). Filtered supernatant containing packaged KFERQ-PS-Dendra vector (Dong et al, 2020) was diluted 1:1 in medium containing 10 µg/mL polybrene (107689, Merck). After 24 h, an additional 1:1 volume was added per well. After 48 h, transduction efficiency was assessed using an inverted widefield fluorescence microscope, the medium was replaced and, if necessary, treatments were added. Cells were carefully washed twice with PBS and fixed using 4% paraformaldehyde (AA433689M, Thermo Fisher) for 15 min at RT, and nuclei were counterstained with 1 µg/mL DAPI in PBS for 5 min. After three 5-minute washes with PBS, samples were mounted using ProLong Diamond and cured overnight. Confocal imaging was performed using a Zeiss LSM 510 (Zeiss) confocal microscope equipped with a ×63 immersion objective.

## Metabolic function assessment

In all experiments, $4 \times 10^4$ cells per well were seeded over Matrigel-coated Seahorse XFe96 plates (103792-100, Agilent). Sensor cartridges were hydrated overnight using Seahorse XF Calibrant (103792-100, Agilent) inside an incubator at 37 °C without CO$_2$. To assess mitochondrial function, cells were washed once and medium wash replaced with XF Base Media (103575-100, Agilent) supplemented with 2 mM glutamine (G8540, Merck), 5.5 mM glucose (G7528, Merck) and 1 mM sodium pyruvate (P5280, Merck) and subjected to sequential injections of 2 µM Oligomycin (11341, Cayman), 1 µM FCCP (15128, Cayman) and 1 µM Antimycin A+Rotenone (34799, 13995, Cayman). To assess

glycolytic capacity, cells were washed once and medium was replaced with XF Base Media supplemented with 2 mM glutamine and 1 mM sodium pyruvate, incubated for 1 h at 37 °C without $CO_2$ and subjected to sequential injections of 20 mM glucose, 2 μM oligomycin, and 100 mM 2-deoxyglucose (14325, Cayman). Oxygen consumption rate (OCR) and extracellular acidification rate (ECAR) were measured using standard MitoStress and GlycoStress protocols using Wave software (v2.6.3.8, Agilent). Recordings were normalized to cell number after addition of 1 μg/mL Hoechst 33342 (H3570, Thermo Fisher) and automated cell number quantification using a Cytation microplate imager (Agilent).

## Lysosomal function assays

In all experiments, $4 \times 10^4$ cells per well were seeded over Matrigel-coated 96-well black clear-bottom plates. A combination of 1 μM Lysotracker Green (lysosomal acidity; L7526, Merck), 1× Magic Red Cathepsin B substrate (acid protease activity; 947, Immuno-Chemistry), and 1 μg/mL Hoechst 33342 was added, and cells were incubated for 15 min at 37 °C, washed, and imaged using a Cytation microplate imager (Agilent).

## Microscopy analyses

### LAMP-2A and HSC70 levels and colocalization
Images were pre-processed by subtracting background with a rolling ball radius of 50 pixels and applying a Gaussian blur filter with a Σ radius of 0.75. The mean fluorescence intensity (MFI) of maximal projections and Manders' M2 coefficient (Bolte and Cordelieres, 2006) in the RPE and neuroretina are reported.

### KFERQ-Dendra and ProteoStat puncta
The number of puncta per cell was quantified in maximal projections using the Find Maxima function with Prominence of >100.

### 4-Hydroxynonenal levels
Images were pre-processed by subtracting the background with a rolling ball radius of 50 pixels and applying a Gaussian blur filter with a Σ radius of 0.75. The MFI of maximal projections of entire cells (>25 per donor) is reported.

### Macroautophagy analysis
Images were pre-processed by subtracting the background with a rolling ball radius of 5 pixels and applying a Gaussian blur filter with a Σ radius of 0.75. The same threshold was applied to each image to obtain p62[+], LC3[+], and LAMP-1[+] masks. Double- and triple-positive masks were obtained using the "AND" function of Image Calculator. Single-, double-, or triple-positive puncta were quantified using 3D Objects Counter (Bolte and Cordelieres, 2006).

### NRF2 nuclear translocation
Cells were manually delineated to create a whole-cell region of interest (ROI) and DAPI staining was used to automatically create a nuclear ROI. MFI was measured in both ROIs, and the ratio of nuclear/cytosolic MFI is reported.

### DNA damage assessment using γH2AX staining
Cells with >3 γH2AX[+] nuclear foci were considered positive, and the percentage of γH2AX[+] cells is reported.

## Bulk and targeted proteomics

For proteomics analyses, $1 \times 10^6$ cells were seeded in Matrigel-coated 6-well plates and maintained as previously described. Treatments were added for 24 h after the final culture medium change. Cells were detached using Accumax, pelleted by centrifugation at $300 \times g$ for 5' and washed twice with sterile ice-cold PBS. After the final centrifugation, PBS wash carefully aspirated using a p1000 micropipette, and the cell pellets were flash-frozen using dry ice. Both targeted and bulk proteomics were performed by the Metabolomics and Proteomics Platform (MAPP) at the Université de Fribourg (Switzerland) as previously described (Leytens et al, 2025), and raw data were processed using Skyline (Pino et al, 2020) and Spectronaut (Biognosys), respectively. Statistical analysis to find differentially enriched proteins was performed using the matrixTests package in R, and GSEA software (Subramanian et al, 2005) was used for pathway enrichment analysis, including 1000 permutations based on gene-set size. CMA literature-validated substrate list was manually curated (Dataset EV2). Canonical and non-canonical (phosphorylation, acetylation) KFERQ-like motif abundance was determined using the KFERQ finder (v0.8) tool (Kirchner et al, 2019). ORA of putative CMA degradation substrates ($FC > 1.25$; $p$ value $< 0.01$; KFERQ-motif[+] in N/L-treated iPSC-RPE) was performed using g:Profiler (Kolberg et al, 2023) including KEGG, Reactome, and WikiPathways databases. $p$ value and fold-change cut-offs were established based on previous proteomics studies from the lab assessing protein abundance differences in RPE and lysosomal degradation of KFERQ proteins (Shang et al, 2024; Bourdenx et al, 2021). Resulting data was exported to CytoScape (v3.10.2.) (Shannon et al, 2003), filtered based on a Jaccard index of 0.25, and clustered using EnrichmentMap (Merico et al, 2010) and AutoAnnotate plugins (Kucera et al, 2016). Heatmaps were generated using the *tidyverse, ggplot2, viridis, RColorBrewer*, and *pheatmap* packages in R (v4.2.0.).

## Graphics

Experimental design diagrams and a graphical abstract were created with BioRender.com.

## Statistics

Data points represent individual donors or donor-derived iPSC-RPE cell lines. Sample size was determined based on previous studies from the lab, including enough cell lines to account for donor heterogeneity and be able to determine significance between healthy/AMD donors. Bioinformatic, biochemical, and microscopy image analyses were performed by a researcher blinded to the condition. All data were evaluated for normality and heteroscedasticity. Normally distributed data were analyzed using a one-way or two-way analysis of variance with appropriate post hoc comparisons (more than two groups) or a two-tailed Student's $t$ test (two groups). Non-normally distributed data were analyzed using Mann–Whitney's $U$-test. All statistical tests were performed using GraphPad Prism (v10.0) or R (v4.2.0), and data are presented as the mean ± standard error of the mean (s.e.m.).

## The paper explained

### Problem

Age-related macular degeneration (AMD) is the leading cause of irreversible central vision loss in elderly people, affecting the macula. As global populations age, the prevalence of AMD is projected to double by 2040. In AMD, the retinal pigment epithelium (RPE) degenerates early on, yet the molecular basis of its selective vulnerability remains poorly understood. We investigated a less well-explored cellular clearance pathway —chaperone-mediated autophagy (CMA)— and its potential role in RPE degeneration in AMD.

### Results

We found that CMA activity is significantly reduced in the RPE of AMD donor eyes, leading to the accumulation of proteins containing KFERQ-like motif-specific tags for CMA degradation. Using AMD patient-derived iPSC-RPE, we confirmed that these cells display impaired CMA, chronic proteotoxic stress, and oxidative damage. Proteomic analysis revealed that both healthy and AMD iPSC-RPE cells rely on CMA to degrade proteins involved in metabolic regulation and signaling. Treatment with the pharmacological CMA activator CA77.1 increased LAMP-2A expression and restored CMA activity. This reduced toxic protein buildup and improved the cellular energy balance of AMD iPSC-RPE.

### Impact

Our findings reveal that the downregulation of CMA is a novel, RPE-specific defect in AMD that contributes to the loss of proteostasis and cellular resilience during disease progression. Pharmacological activation of CMA alleviated multiple hallmarks of RPE degeneration in patient-derived cells. These results open new therapeutic avenues for treating AMD by targeting CMA to enhance cellular clearance, restore metabolic function, and protect the RPE. Modulating this pathway could be a strategy for preventing or slowing vision loss in AMD and possibly other retinal degenerations.

## Data availability

Newly generated bulk and targeted proteomic data from iPSC-RPE have been deposited to ProteomeXchange database under accession code (PXD051893; https://www.ebi.ac.uk/pride/archive/projects/PXD051893). Public transcriptomic and proteomic datasets used for this study are summarized in Dataset EV1.

The source data of this paper are collected in the following database record: biostudies:S-SCDT-10_1038-S44321-025-00329-w.

## Peer review information

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

## Acknowledgements

We thank the Microscopy Facility at Doheny Eye Institute; Jorge Montesinos for providing reagents; M. Stumpe and A. Leytens for proteomics support. The authors thank all other members of the Ferrington and Autophagy Lab for thoughtful discussions and support. Research in the P.B. lab is supported by grants 310030_215271 from the Swiss National Science Foundation (SNSF) and PID2021-126864NBI00 from MCIN, Spain. JIJ-L is supported by an FPI predoctoral fellowship (PRE2019-088222) from MCIN, Spain. Research in the DAF and JRD laboratories is supported by a grant from the National Institutes of Health/ National Eye Institute R01EY028554. Research in the AMC lab is supported by grants from the National Institutes of Health/National Institute on Aging AG021904 and AG031782 and the generous support of the JPB Foundation and Hevolution Foundation. AM-S was supported by a Ramón Areces Postdoctoral Fellowship. Research in the JD lab is supported by the Canton and the University of Fribourg as part of the SKINTEGRITY.CH collaborative research project and by the Swiss National Science Foundation (grants 310030_212187 and CRSII5_189952 to JD).

## Author contributions

**Juan Ignacio Jiménez-Loygorri**: Conceptualization; Data curation; Formal analysis; Validation; Investigation; Visualization; Methodology; Writing—original draft; Writing—review and editing. **Peng Shang**: Investigation; Methodology; Writing—review and editing. **Ibrahim Bayramoglu**: Formal analysis; Investigation; Methodology; Writing—review and editing. **Raquel Gómez-Sintes**: Methodology; Writing—review and editing. **Adrián Martín-Segura**: Investigation; Writing—review and editing. **Helena Ambrosino**: Investigation; Methodology; Writing—review and editing. **Johnson Hoang**: Investigation; Methodology; Writing—review and editing. **Antonio Díaz**: Investigation; Writing—review and editing. **Zhaohui Geng**: Methodology; Writing—review and editing. **Evripidis Gavathiotis**: Resources; Methodology; Writing—review and editing. **James R Dutton**: Resources; Methodology; Writing—review and editing. **Jörn Dengjel**: Resources; Methodology; Writing—review and editing. **Ana Maria Cuervo**: Resources; Methodology; Writing—review and editing. **Deborah A Ferrington**: Conceptualization; Resources; Funding acquisition; Methodology; Project administration; Writing—review and editing. **Patricia Boya**: Conceptualization; Supervision; Funding acquisition; Methodology; Writing—original draft; Project administration; Writing—review and editing.

Source data underlying figure panels in this paper may have individual authorship assigned. Where available, figure panel/source data authorship is listed in the following database record: biostudies:S-SCDT-10_1038-S44321-025-00329-w.

## Disclosure and competing interests statement

AMC and EG are co-founders and scientific advisors for the autophagy program at Life Biosciences. The other authors declare no competing interests.

# Expanded View Figures

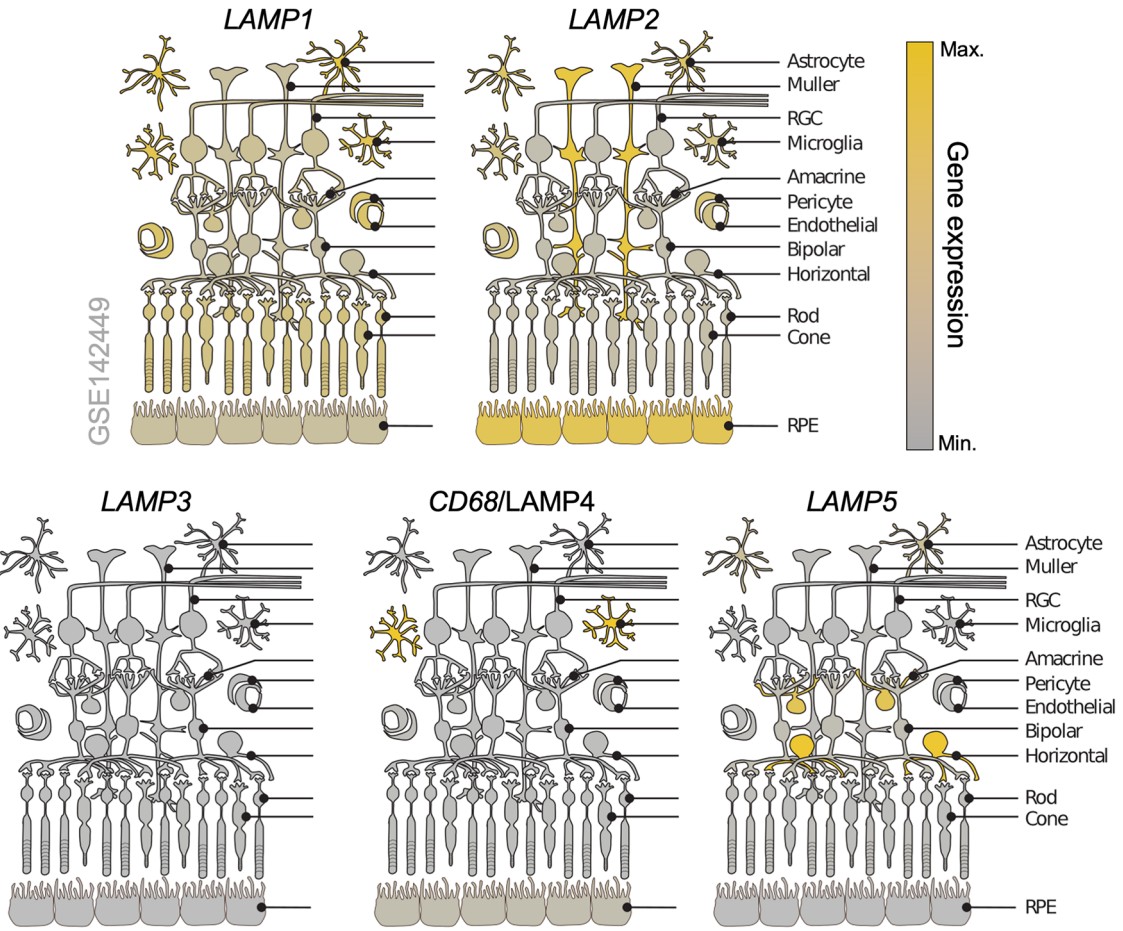

**Figure EV1.  *LAMP2* is preferentially expressed in the RPE and glial cells.**

Visual heatmap of mRNA levels of different LAMPs in the neuroretina and RPE, generated using *Spectacle* and publicly available scRNA-seq data (GSE142449).

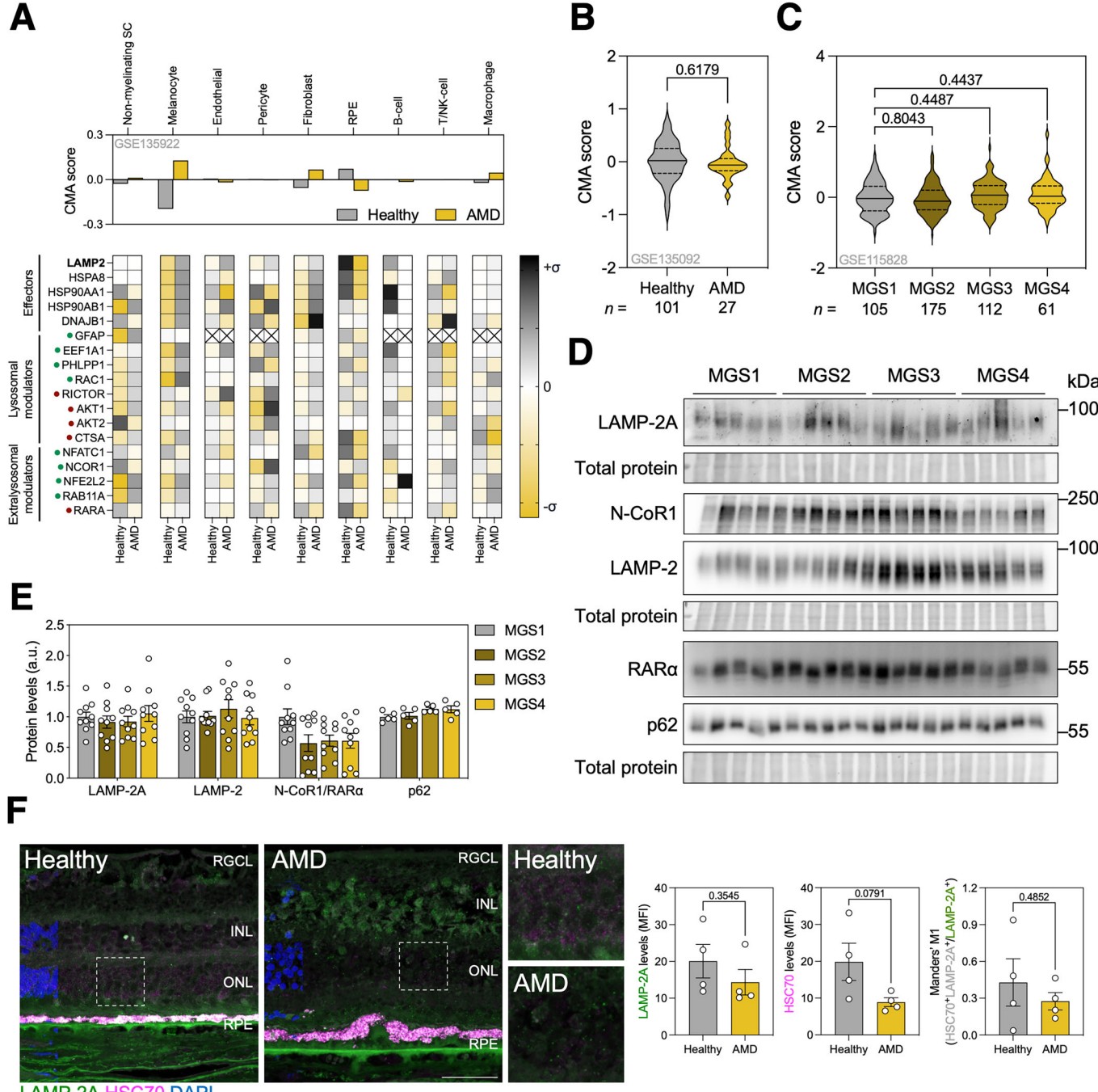

**Figure EV2. CMA is not affected in other cell types within the RPE/choroid interface or the neuroretina.**

(A) Heatmap showing the mRNA levels of the CMA network components (bottom) and CMA activation score obtained from a publicly available RNA-seq dataset (GSE135922; top). (B) CMA activation score in the macular neuroretina obtained from a publicly available RNA-seq dataset (GSE135092). (C) CMA activation score in the neuroretina of MGS-graded donors obtained from a publicly available scRNA-seq dataset (GSE115828). (D) Western blot analysis of CMA-related proteins (LAMP-2A, N-CoR1, RARα, LAMP-2) and macroautophagic substrates (p62) in the neuroretina of MGS-graded donors. (E) Quantification as shown in (D). ($n = 5$–10). (F) Representative images and quantification of donor sections immunostained against LAMP-2A (green) and HSC70 (magenta), nuclei were counterstained with DAPI (blue). Quantification of the levels of both proteins (MFI) and the proportion of lysHSC70 CMA-proficient lysosomes in the neuroretina is shown. ($n = 4$). Scale bar, 50 μm. All data are expressed as the mean ± s.e.m. Dots represent individual donors. $p$ values were calculated using unpaired Student's $t$ test ((B), F (LAMP-2A, HSC70)), Kruskal–Wallis with Dunn's post hoc test (C) or Mann–Whitney's $U$-test ((F) (Manders' M1)). Source data are available online for this figure.

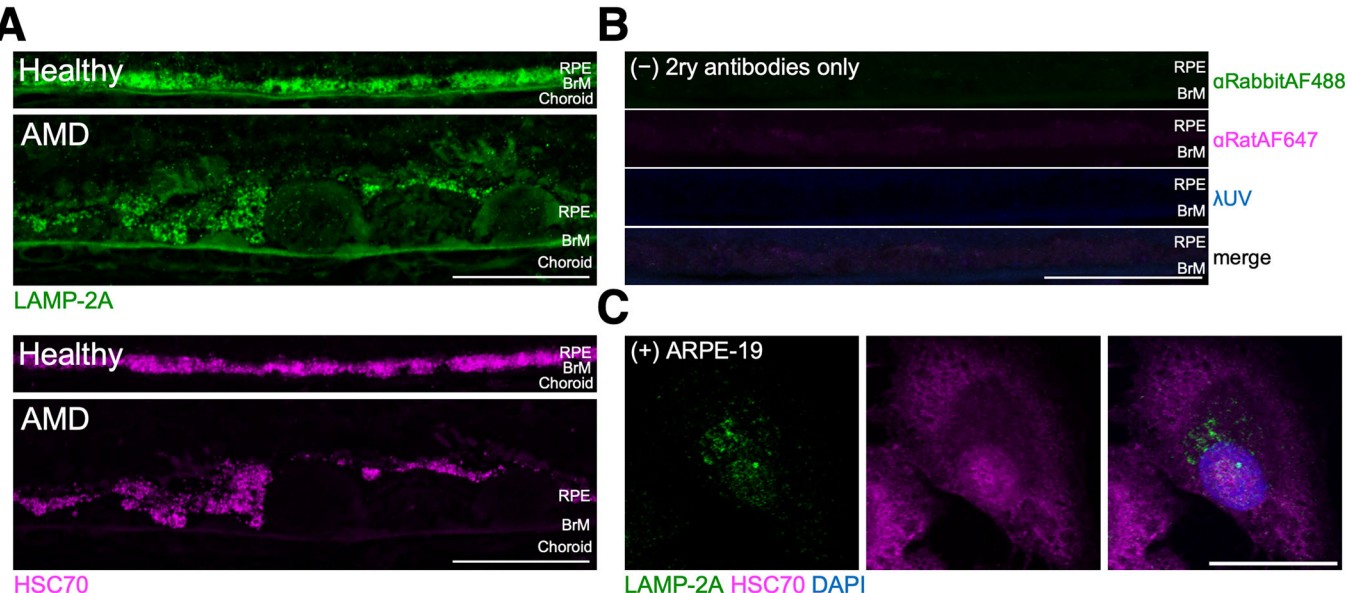

**Figure EV3. LAMP-2A and HSC70 immunofluorescence in the RPE of healthy and AMD donors.**

(A) Single-channel images of LAMP-2A (top, green) and HSC70 (bottom, magenta) immunostaining of donor eyes as shown in Fig. 1E. (B) Negative secondary antibody-only control. (C) ARPE-19 human cells immunostained against LAMP-2A (green) and HSC70 (magenta), nuclei were counterstained with DAPI (blue). Scale bars, 50 (A, B) and 25 (C) μm.

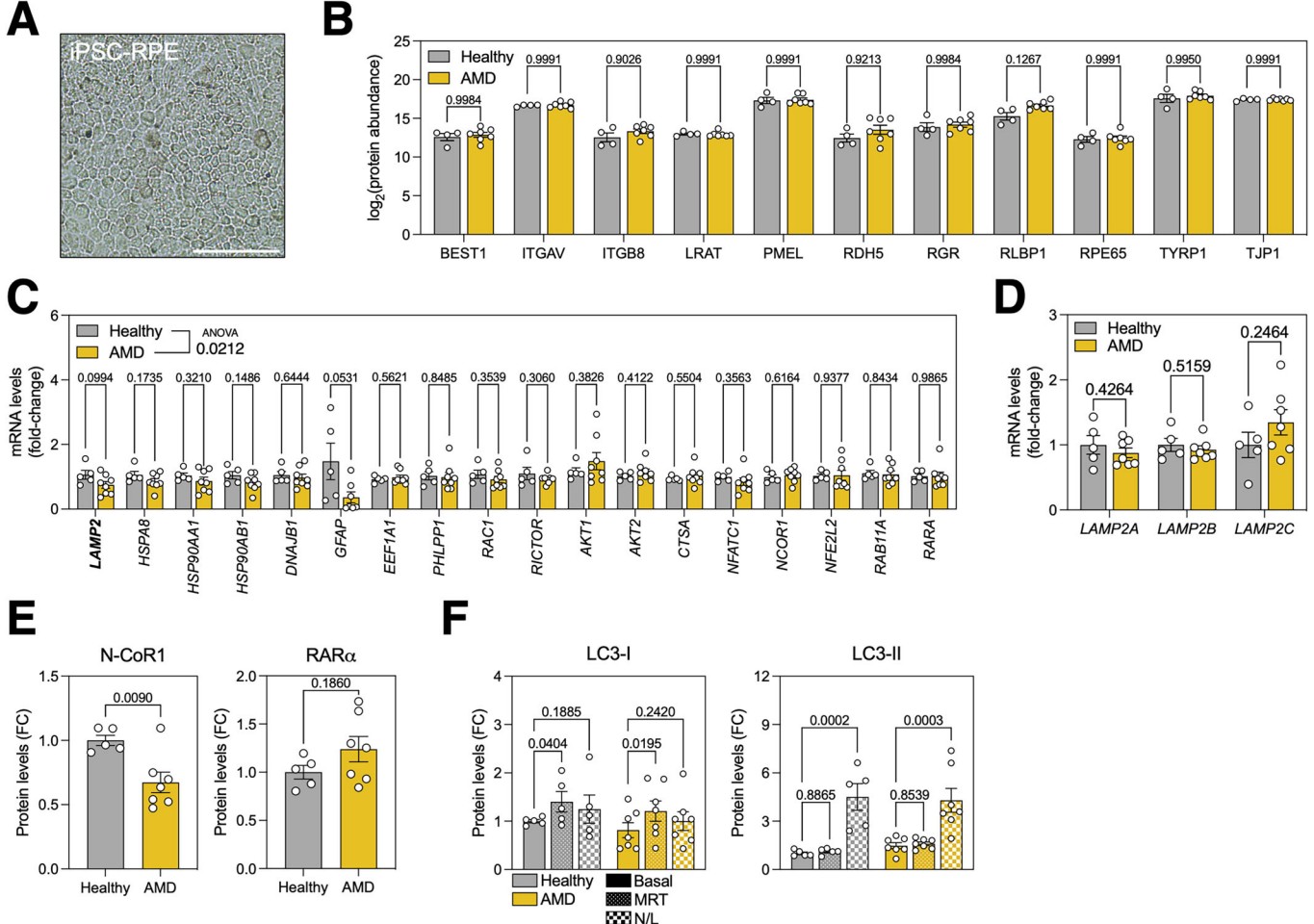

**Figure EV4. Expression levels of CMA network components in iPSC-RPE.**

(A) Representative bright field image of differentiated iPSC-RPE at P3. (B) Validation of RPE identity by measuring the levels of prototypic RPE proteins by bulk proteomics. ($n = 4–7$). (C) mRNA levels of CMA network components used for CMA activation score calculation in iPSC-RPE, as shown in Fig. 2B. ($n = 5–8$). (D) mRNA levels of the different LAMP2 splicing variants in iPSC-RPE. ($n = 5–7$). (E) Quantification of protein levels of N-CoR1 and RARα, as shown in Fig. 2C. ($n = 5–7$). (F) Quantification of protein levels of LC3-I and LC3-II, as shown in Fig. 3E. ($n = 5–7$). Scale bar, 100 μm. All data are expressed as the mean ± s.e.m. Dots represent individual donors. p values were calculated using unpaired Student's t test (B–E) or two-way ANOVA (F). Source data are available online for this figure.

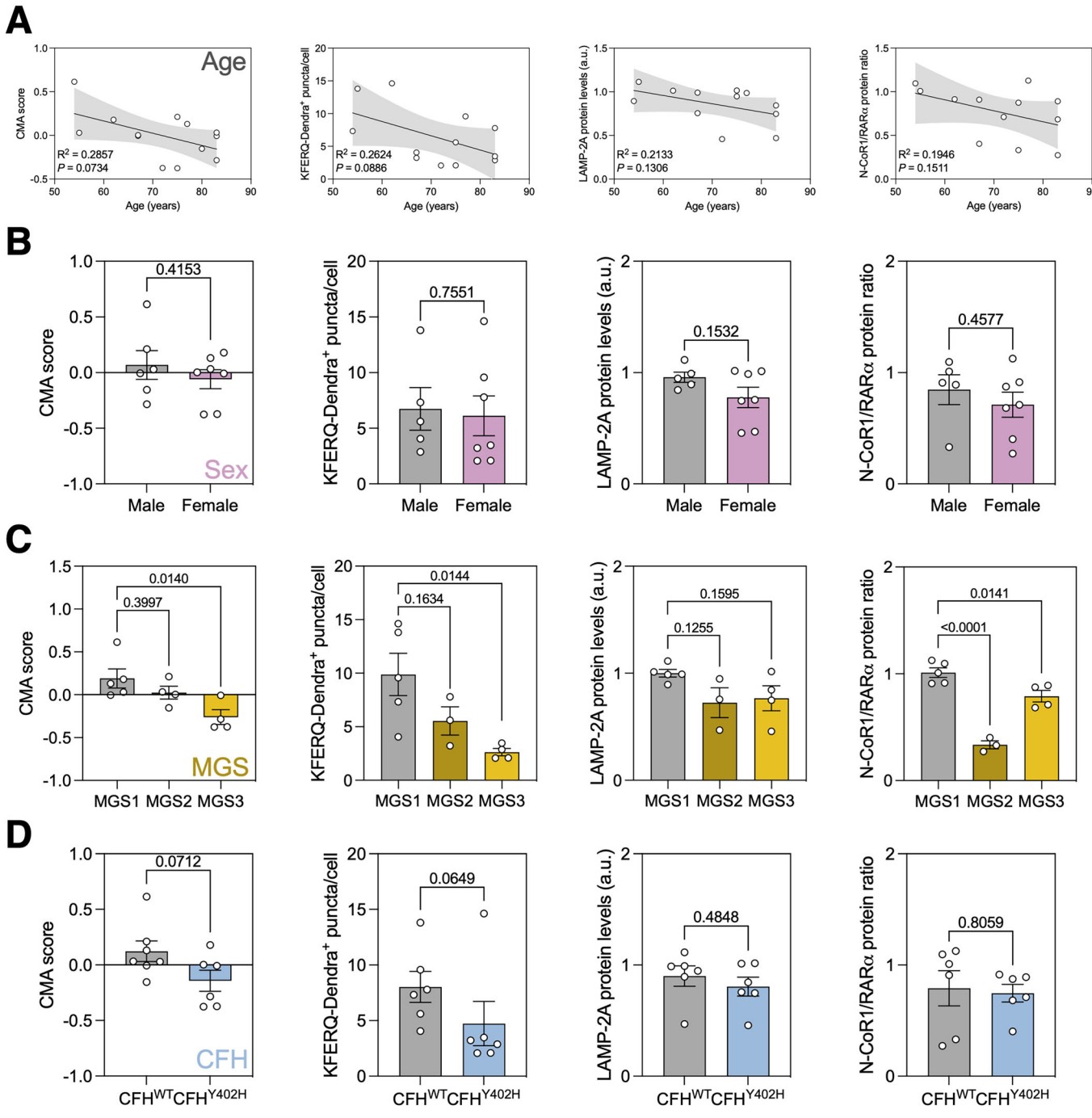

**Figure EV5.  CMA activity in the RPE is impacted by age and AMD severity.**

Analysis of CMA activation, KFERQ-Dendra+ puncta/cell, LAMP-2A protein levels and N-CoR1/RARα ratio in iPSC-RPE classified according to donor's age (**A**), sex (**B**), MGS grading (**C**), and presence of the high-risk CFH[Y402H] variant (**D**). ($n = 4$–12). All data are expressed as the mean ± s.e.m. Dots represent individual donors. $p$ values were calculated using simple linear regression (**A**), unpaired Student's $t$ test (**B** [CMA score, LAMP-2A, N-CoR1/RARα], **D** [CMA score, N-CoR1/RARα]), Mann–Whitney's $U$-test (**B** [KFERQ-Dendra] **D** [KFERQ-Dendra, LAMP-2A]), or one-way ANOVA with Dunnett's post hoc test (**C**). Source data are available online for this figure.

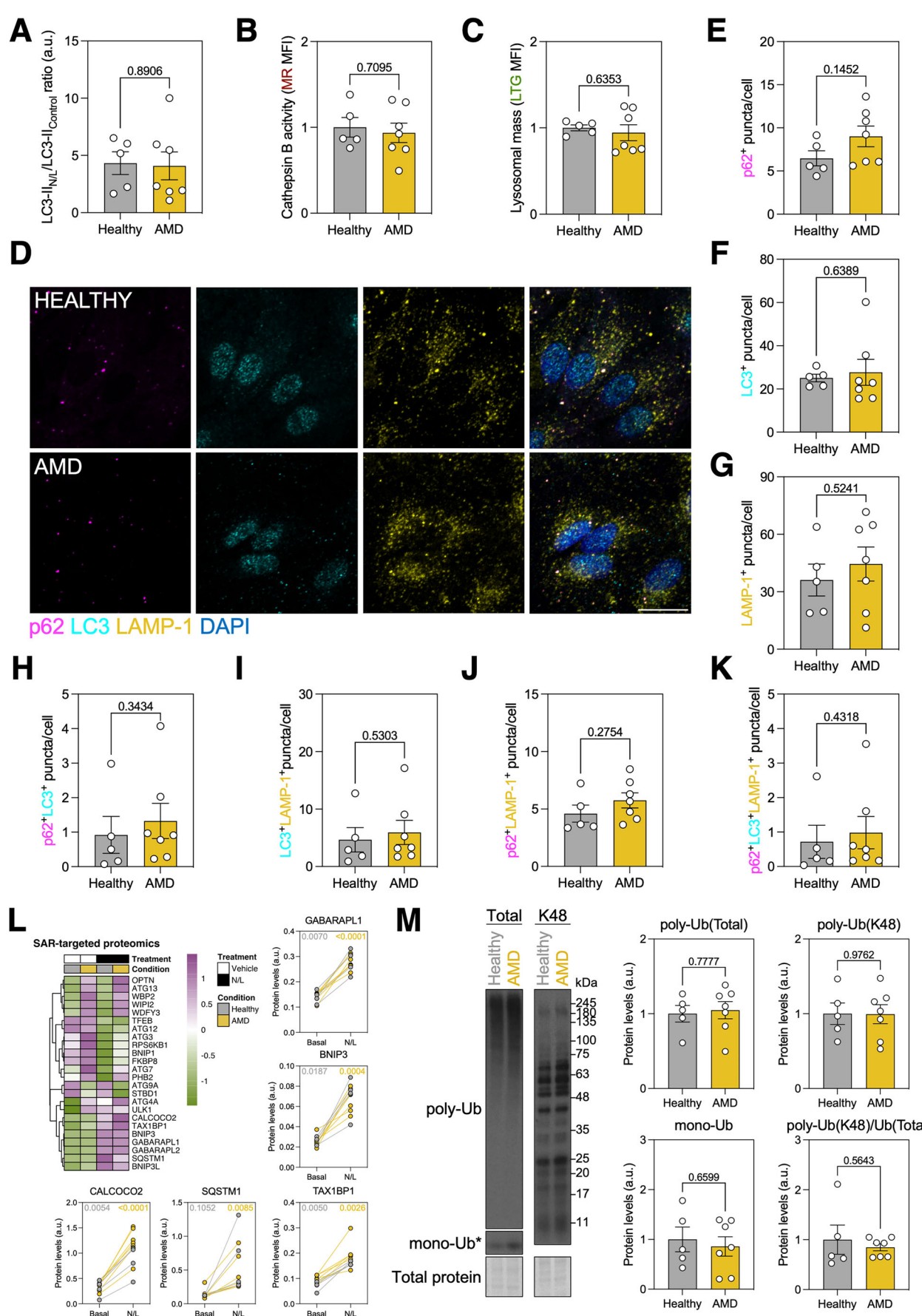

◀ **Figure EV6. Macroautophagy is functional in both healthy and AMD iPSC-RPE.**

(A) Quantification of autophagic flux (LC3-II$_{N/L}$/LC3-II$_{Basal}$) as shown in Fig. 3E. ($n = 5$–7). (B) Quantification of cathepsin B proteolytic activity using Magic Red fluorogenic substrate, analyzed by live imaging. ($n = 5$–7). (C) Quantification of acidic lysosomal mass using LysoTracker Green, analyzed by live imaging. ($n = 5$–7). (D) Immunostaining analysis of p62$^+$ (magenta; macroautophagic cargo), LC3$^+$ (cyan; autophagosomes) and LAMP-1$^+$ (yellow; lysosomes) vesicles in iPSC-RPE, nuclei were counterstained with DAPI (blue). (E–K) Quantification of the number of single-, double-, and triple-positive puncta per cell as shown in D. ($n = 5$–7). (L) Heatmap and representative graphs of selective autophagy receptor (SAR) targeted proteomics of iPSC-RPE treated for 24 h with N/L. ($n = 4$–7). (M) Western blot analysis of the levels of total mono-ubiquitin, total poly-ubiquitinated (left), and K48-linked poly-ubiquitinated (right) proteins. ($n = 5$–7). Scale bar, 25 µm. All data are expressed as the mean ± s.e.m. Dots represent individual donors. $p$ values were calculated using unpaired Student's $t$ test (A–C, E, G, J, M), Mann–Whitney's $U$-test (F, H, I, K), or paired Student's $t$ test (L). Source data are available online for this figure.

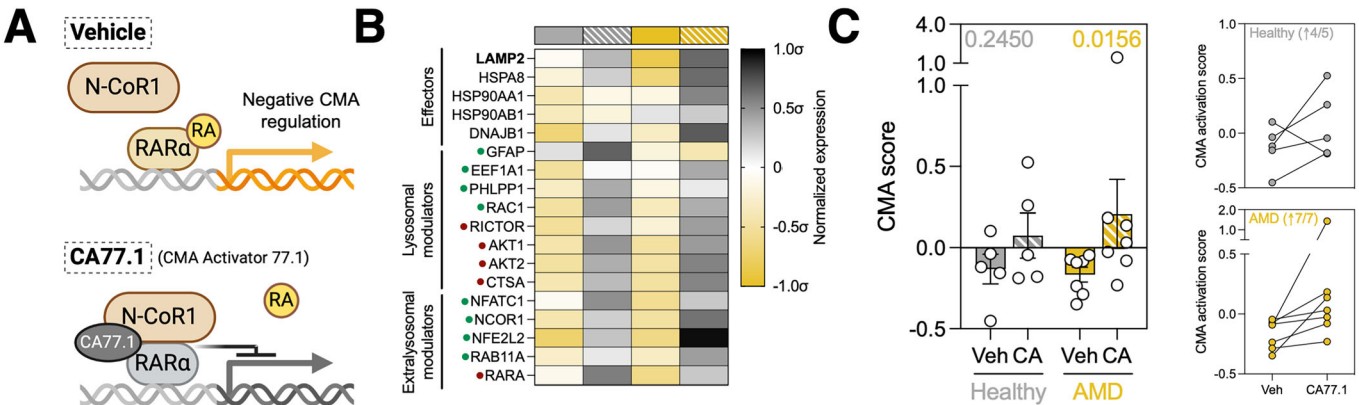

**Figure EV7. CMA activator CA77.1 induces CMA in iPSC-RPE.**

(A) Diagram depicting the mechanism of action of CA77.1, which, by stabilizing the interaction between N-CoR1 and RARα, downregulates a subset of transcriptional retinoic acid (RA) signaling involved in CMA inhibition. (B) Heatmap showing the mRNA levels of the CMA network components in iPSC-RPE treated with 10 μM CA77.1 for 24 h. (C) CMA activation score in iPSC-RPE treated with CA77.1. ($n = 5$–7). All data are expressed as the mean ± s.e.m. Dots represent individual donors. $p$ values were calculated using paired Student's $t$ test. Source data are available online for this figure.

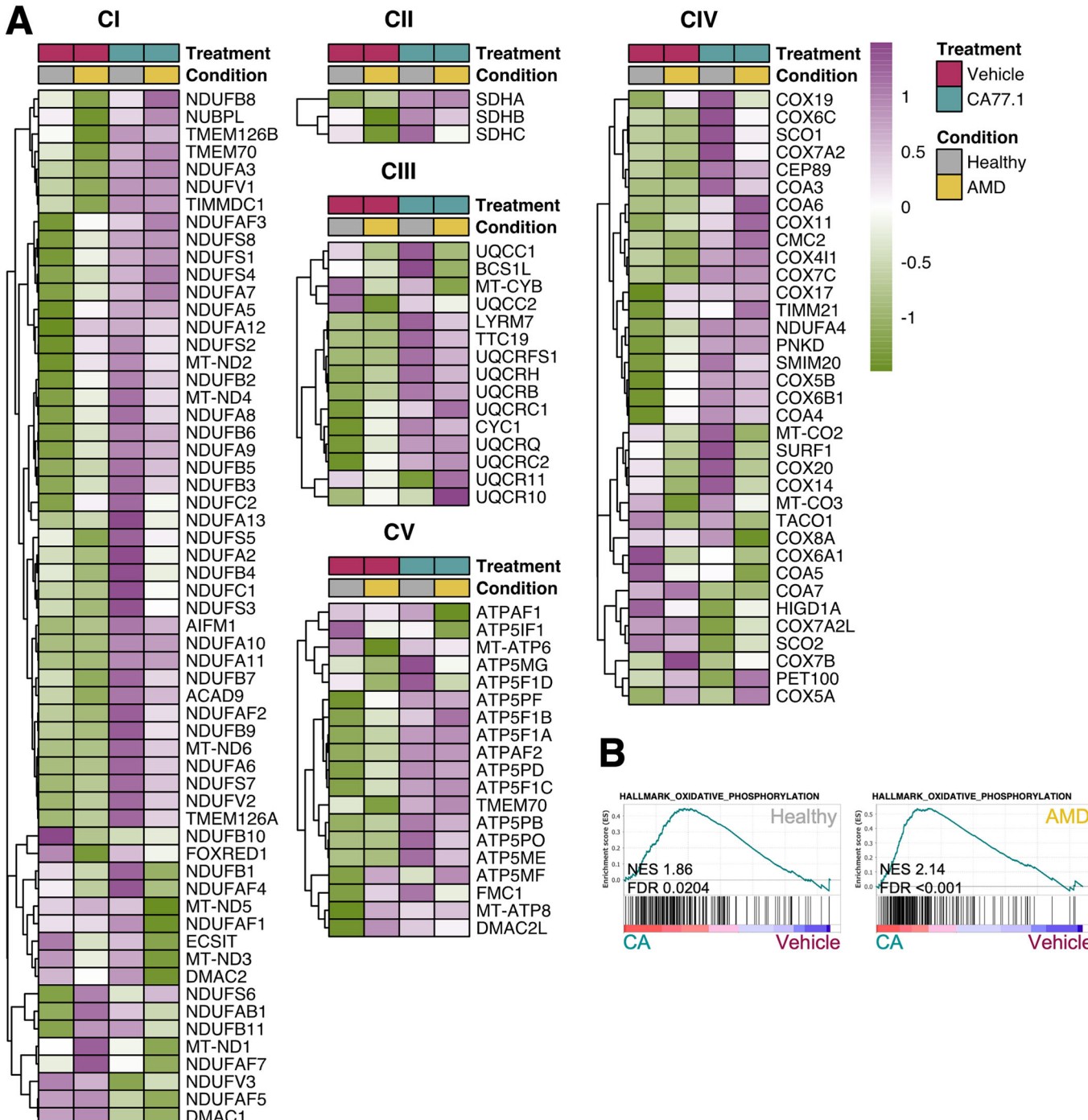

**Figure EV8. CA77.1 induces mitochondrial biogenesis.**

(A) Heatmaps showing the protein levels of all detected components of mitochondrial complexes (CI–V) of the ETC in iPSC-RPE treated with 10 μM CA77.1 for 24 h. (B) Upregulation of Hallmark Oxidative phosphorylation pathway in healthy and AMD iPSC-RPE analyzed by GSEA.

