## [Peer Review File · EMBO Molecular Medicine]

Defective chaperone-mediated autophagy in the retinal pigment epithelium of age-related macular degeneration patients

Juan Ignacio Jiménez-Loygorri, Peng Shang, Ibrahim Bayramoglu, Raquel Gomez Sintes, Adrián Martín-Segura, Helena Ambrosino, Johnson Hoang, Antonio Diaz, Zhaohui Geng, Evripidis Gavathiotis, James Dutton, Jörn Dengjel, Ana Cuervo, Deborah Ferrington, and Patricia Boya

Corresponding authors: Juan Ignacio Jiménez-Loygorri (jjimenezl@cni.es) , Patricia Boya (patricia.boya@unifr.ch), Deborah Ferrington (DFerrington@doheny.org)

Review Timeline:

Submission Date:	6th Apr 25
Editorial Decision:	5th May 25
Revision Received:	25th Aug 25
Editorial Decision:	18th Sep 25
Revision Received:	6th Oct 25
Accepted:	13th Oct 25

Editor: Lise Roth

Transaction Report:

5th May 2025

Dear Dr. Jiménez-Loygorri,

Thank you for the submission of your manuscript to EMBO Molecular Medicine. We have now received feedback from the three reviewers who agreed to evaluate your manuscript. As you will see from the reports below, the referees acknowledge the interest of the study and are overall supporting publication of your work pending appropriate revisions.

Addressing the reviewers' concerns in full will be necessary for further considering the manuscript in our journal, and acceptance of the manuscript will entail a second round of review. However, after further discussion within the team and with the referees, we agreed that the *in vivo* validation mentioned by referee #2 will NOT be requested for further consideration. Instead, please discuss this limitation in the text.

EMBO Molecular Medicine encourages a single round of revision only and therefore, acceptance or rejection of the manuscript will depend on the completeness of your responses included in the next, final version of the manuscript. For this reason, and to save you frustration at the end, I would strongly discourage you from returning an incomplete revision.

We are expecting your revised manuscript within three to four months, if you anticipate any delay, please contact us.

We require:

4) A .docx formatted letter INCLUDING the reviewers' reports and your detailed point-by-point responses to their comments. As part of the EMBO Press transparent editorial process, the point-by-point response is part of the Review Process File (RPF), which will be published alongside your paper.

5) A complete author checklist, which you can download from our author guidelines (<https://www.embopress.org/page/journal/17574684/authorguide#submissionofrevisions>). Please insert information in the checklist that is also reflected in the manuscript. The completed author checklist will also be part of the RPF.

6) All Materials and Methods need to be described in the main text using our 'Structured Methods' format. According to this format, the Methods section includes a Reagents and Tools Table (listing key reagents, experimental models, software and relevant equipment and including their sources and relevant identifiers) followed by a Methods and Protocols section describing the methods, ideally using a step-by-step protocol format. The aim is to facilitate adoption of the methodologies across labs. Please download and fill our Reagents and Tools Table template (.docx), which you can find in our author guidelines: <https://www.embopress.org/page/journal/14693178/authorguide#structuredmethods>.

7) Please note that all corresponding authors are required to supply an ORCID ID for their name upon submission of a revised manuscript.

8) It is mandatory to include a 'Data Availability' section after the Materials and Methods. Before submitting your revision, primary datasets produced in this study need to be deposited in an appropriate public database, and the accession numbers and database listed under 'Data Availability'. Please remember to provide a reviewer password if the datasets are not yet public (see <https://www.embopress.org/page/journal/17574684/authorguide#dataavailability>).

9) For data quantification: please specify the name of the statistical test used to generate error bars and P values, the number (n) of independent experiments (specify technical or biological replicates) underlying each data point and the test used to calculate p-values in each figure legend. The figure legends should contain a basic description of n, P and the test applied. Graphs must include a description of the bars and the error bars (s.d., s.e.m.). Please provide exact p values.

10) Our journal encourages inclusion of *data citations in the reference list* to directly cite datasets that were re-used and obtained from public databases. Data citations in the article text are distinct from normal bibliographical citations and should directly link to the database records from which the data can be accessed. In the main text, data citations are formatted as follows: "Data ref: Smith et al, 2001" or "Data ref: NCBI Sequence Read Archive PRJNA342805, 2017". In the Reference list, data citations must be labeled with "[DATASET]". A data reference must provide the database name, accession number/identifiers and a resolvable link to the landing page from which the data can be accessed at the end of the reference. Further instructions are available at .

11) We replaced Supplementary Information with Expanded View (EV) Figures and Tables that are collapsible/expandable online. EV Figures should be cited as 'Figure EV1, Figure EV2' etc... in the text and their respective legends should be included in the main text after the legends of regular figures.

12) The paper explained: EMBO Molecular Medicine articles are accompanied by a summary of the articles to emphasize the major findings in the paper and their medical implications for the non-specialist reader. Please provide a draft summary of your article highlighting

13) Author contributions: CRedit has replaced the traditional author contributions section because it offers a systematic machine readable author contributions format that allows for more effective research assessment. Please remove the Authors Contributions from the manuscript and use the free text boxes beneath each contributing author's name in our system to add specific details on the author's contribution. More information is available in our guide to authors.

Please also suggest a visual abstract to illustrate your article as a PNG file 550 px wide x 300-600 px high. A cropped portion of this image will serve as thumbnail for the table of content on our webpage.

16) As part of the EMBO Publications transparent editorial process initiative (see our Editorial at <http://embomolmed.embopress.org/content/2/9/329>), EMBO Molecular Medicine will publish online a Review Process File (RPF) to accompany accepted manuscripts.

In the event of acceptance, this file will be published in conjunction with your paper and will include the anonymous referee reports, your point-by-point response and all pertinent correspondence relating to the manuscript. Let us know whether you agree with the publication of the RPF and as here, if you want to remove or not any figures from it prior to publication. Please note that the Authors checklist will be published at the end of the RPF.

I look forward to receiving your revised manuscript.

Yours sincerely,

Lise Roth

***** Reviewer's comments *****

Referee #1 (Remarks for Author):

The authors want to study chaperone-mediated autophagy (CMA) in RPE from healthy or AMD (age macular degeneration) patients. CMA is a selective type of autophagy for protein containing a specific motif: KFERQ-like motif. They compare healthy and AMD donor samples and donor-derived iPSC-RPE lines. And they used a combination of techniques to analyse gene and protein expression and functional metabolic assays. They show that CMA activity is globally decrease in RPE from AMD patients and AMD-iPSC compare to healthy ones. Moreover, treatment with a CMA activator (CA77.1) restores several hallmarks of CMA dysfunction in patients-RPE.

This paper is of interest for the community but needs some clarifications before publications.

Introduction:

- Line 69, authors say that "no treatment is currently available for the dry subclass of disease". And the next sentence says that some treatments target the complement pathway have recently been approved...Maybe it is better to rephrase and suppress the "no treatment...".

- Line 113: add "conjunctiva-derived epithelial cells..."

Results:

- Line 125: precise what stage of AMD patients. In a more general way, it would be nice to have the stage of AMD patients for sample coming directly from patients but also the ages of patients donating their conjunctiva-epithelial cells. The choice of conjunctiva should be explained.

- Line 136: a need for higher magnification for fig 1e, arrowheads is needed, as we do not see the green dots.

- Line 141: the role and interest of studying N-CoR1/RAR ratio need to be introduced.

- Line 146: I do not understand why the authors are stating that they observed "a coetaneous impairment of CMA in the RPE...". Coetaneous with what? They might want to add a sentence that summarize all the deregulated processes seen in AMD patients samples.

- Line 156, in iPSC-derived RPE they observe decrease LAMP-2A proteins without changes in mRNA levels. They should comment on that as it is also different from what it is observed in patients samples?

Globally, results would benefit from clarifications as it is not always very easy to follow all the hallmarks that are deregulated or not.

Methods:

- Line 384: the protocol used for the obtention of iPSC and then the RPE should be re-written even if it is coming from a previous publication, with a separation between the obtention of iPSC and the differentiation of RPE cells.

- Line 410: FFPE slides should be explained. Maybe the paragraphs of immunohistochemistry and immunocytochemistry could be reunited?

- Line 461: the process of using lentiviruses should be explained.

- Line 474: the term BGT should be explained.

Referee #2 (Comments on Novelty/Model System for Author):

The study provides novel insight on the role of a specific arm of autophagy pathways in dry age-related macular degeneration. While the findings are generally interesting, they could be improved by demonstration of selectivity of their tool compound CA77.1 as well as demonstration of clinical relevance in models of disease.

Referee #2 (Remarks for Author):

In the study "Defective chaperone-mediated autophagy in the retinal pigment epithelium of age-related macular degeneration patients" by Shang et al, the authors provide a series of data from human and conjunctiva-derived iPSC of RPE that demonstrates a general reduction of chaperon-mediated autophagy (CMA) (one of the 3 autophagy pathways that is specific for proteins containing KFERQ-like motifs) in patients with AMD. The authors further show that treatment with an activator of CMA reestablishes proteostasis and reestablishes RPE proteomes in AMD donor RPE. They propose that CMA activators can be explored therapeutically for AMD.

The manuscript is well written, and study well executed. I have the following points for amelioration. Specifically with regards to the use of CMA stimulators and their clinical relevance.

-Figure 1e: The claim for extracellular vesicles in Drusen need substantiation. A higher magnification image would be desirable and evidence that the corresponding vesicles are indeed exocytosed. Drusen may have cells in them so a counter stain would help.

- I would suggest working in the table of patient baseline characteristics for the iPSC experiments, into the main manuscript.

- The authors describe that healthy iPSC-RPE showed preferential lysosomal 200 degradation of proteins involved in glycolysis and metabolic adaptation, while AMD 201 iPSC-RPE showed lysosomal degradation of proteins involved in fatty acid metabolism, 202 ferroptosis and ephrin signaling (Figure 3c). Can the authors elaborate here what this potential switch in effectors of energy metabolism means?

-The authors use CA77.1 and a concentration of 10uM. This is quite elevated. What data can the authors present on the specificity and pharmacodynamic properties of the compound given this elevated dose? An increase in NRF2 may actually be a stress response to the compound.

- I found the labeling of figure 5 hard to follow with respect to what groups were compound stimulated. Please provide legends throughout.

- Any attempts for in vivo validation of concepts presented in models of disease?

Referee #3 (Comments on Novelty/Model System for Author):

Both models and methodologies are state-of-the-art
the study may lead to novel pharmacological approaches in AMD.

Referee #3 (Remarks for Author):

Loss of autophagy pathways is linked to several, if not all, age-related diseases. In this manuscript, the authors report that the activity of chaperone-mediated autophagy (CMA) declines in the retinal pigment epithelium (RPE) in age-related macular degeneration patients, compared with healthy age-matched individuals. Previous studies had pointed out decreased macroautophagy, lysosomal and mitochondrial dysfunctions as potential causative factors in AMD, but whether LAMP-2A lysosomes specifically involved in CMA contributed to AMD pathology remained elusive.

Using elegant orthogonal approaches in AMD donor samples and iPSC-RPE lines derived from donor's conjunctiva, the authors uncovered that defective CMA is a major contributor to RPE dysfunctions. Specifically, they found that CMA is responsible for the proteostasis of a selective subsets of KFERQ-containing proteins, involved in lipid metabolism among others pathways. Importantly, pharmacological treatment with a CMA activator rescued the affected proteome of the cells from AMD donors and AMD iPSC-RPE, an effect attributed to lowering oxidative stress, through the upregulation of NRF2, and improved mitochondrial function.

Altogether, the manuscript presents some very interesting and clinically relevant data elucidating a mechanism by which CMA governs the healthy state of RPE and identifies CMA as an actionable target in AMD.

The experimental design and the quality of the data are robust and support the conclusions of the study. A few points need to be further addressed by the authors to clarify some findings and/or further strengthen the main conclusions of the manuscript.

Main comments:

1) Figure 1b, is the CMA score significantly reduced in the RPE of patients with AMD? Please be careful to describe results that do not reach statistical significance as a trend, and remove the word "significant/ly" from the descriptions of the data.

2) Under serum starvation CMA activity is increased (figure 2f), but most likely also macroautophagy is under these conditions. What is the relative contribution of these autophagy pathways to the AMD phenotype? Is the ProteoStat+ staining as a proxy of CMA, also rescued by starvation?

- 3) Data on perturbed glutathione biosynthesis in AMD are interesting. Can the authors actually measure the levels of GSH in their iPSC-RPE lines? Please note that 4-hydroxynonenal (4-HNE) is a byproduct of lipid-peroxidation.
- 4) Can the author explain in the text the selection criteria, based on the fold-change ($FC > 1.25$), p-value ($p < 0.01$) to filter the number of enriched protein hits in healthy and AMD conditions?
- 5) In the western blot of Figure 3e basal levels of LAMP-2A are not different from healthy and AMD derived iPSC-RPE? Also from the representative WB ACSL4 levels are markedly elevated in basal conditions in AMD when compared with Healthy donors, but this is not reflected in the quantifications in panel 3f. Are there other ferroptosis-related proteins that accumulate as CMA declines in AMD, which could be contributing to the increased level of oxidative stress observed under this pathological condition? And related to this, have the authors observed ferroptosis as a contributing factor in the degeneration of AMD-derived iPSC-REP cell lines?
- 6) Can the authors discuss further how the lysosomal degradation of KFERQ-motif containing proteins affects cholesterol metabolism? Do the AMD-derived iPSC-RPE cells have problems in maintaining cholesterol fluxes (uptake and efflux)? Is cholesterol accumulated in the lysosomes of AMD-derived cells?
- 7) In the schematic panel of Fig 3d the ULK1/2 inhibitor MRT68921 seems to block degradation of the autolysosome, whereas it prevents autophagosome formation. In line with this it does not increase LC3-II accumulation as shown in WB of Fig. 3e (it should cause an accumulation of the LC3-I form).
- 8) Data using the CMA activator is captivating. Mechanistically does CA77.1 increase the expression levels and nuclear translocation of NRF2 in AMD-derived RPE by targeting Keap1? Is the enhanced NRF2 activity responsible for LAMP-2A expression after CA77.1 treatment? This could indicate a feedforward positive loop. Please also consider to increase the quality of Fig 4b, nuclear immunostaining of NRF2 is barely detectable.

******* Reviewer's comments *******

We thank the editor and reviewers for their insightful and thorough review of our manuscript that has substantially improved it. Text changes are shown in **blue**, and figures have been updated accordingly.

Referee #1 (Remarks for Author):

The authors want to study chaperone-mediated autophagy (CMA) in RPE from healthy or AMD (age macular degeneration) patients. CMA is a selective type of autophagy for protein containing a specific motif: KFERQ-like motif. They compare healthy and AMD donor samples and donor-derived iPSC-RPE lines. And they used a combination of techniques to analyse gene and protein expression and functional metabolic assays. They show that CMA activity is globally decrease in RPE from AMD patients and AMD-iPSC compare to healthy ones. Moreover, treatment with a CMA activator (CA77.1) restores several hallmarks of CMA dysfunction in patients-RPE. This paper is of interest for the community but needs some clarifications before publications.

We thank the reviewer for their feedback that has helped increase the quality and scientific impact of our manuscript.

Introduction:

- Line 69, authors say that "no treatment is currently available for the dry subclass of disease". And the next sentence says that some treatments target the complement pathway have recently been approved...Maybe it is better to rephrase and suppress the "no treatment...".

We agree with the reviewer, we have now clarified the effectiveness of treatments available for both wet and dry AMD:

"Antiangiogenic immunotherapy (anti-VEGF) has been shown to effectively slow the progression of wet AMD(Deng et al., 2022). For dry AMD, treatments that target the complement pathway have shown a moderate improvement and have recently been approved for clinical use, but their long-term efficacy remains to be fully characterized(Heier et al, 2023; Khanani et al, 2023)."

- Line 113: add "conjunctiva-derived epithelial cells..."

Text has been updated accordingly.

Results:

- Line 125: precise what stage of AMD patients. In a more general way, it would be nice to have the stage of AMD patients for sample coming directly from patients but also the ages of patients donating their conjunctiva-epithelial cells. The choice of conjunctiva should be explained.

We agree on the importance of sharing patient demographics, we have now included all additional information available on age, sex, AMD staging and available medical records for omics datasets analyzed in new **Table EV1**. Donor data for

immunohistochemistry analysis and iPSC-RPE is also provided in **Table EV3** and **EV4**, respectively. Additionally, and as per suggestion of **Referee #2**, key information (age, sex, risk SNP genotyping and AMD staging) about iPSC-RPE cell lines is provided in new **Table 1**.

- Line 136: a need for higher magnification for fig 1e, arrowheads is needed, as we do not see the green dots.

We apologize for the low resolution of microscopy images as a result of file conversion during manuscript preparation. Additionally, single channel images for LAMP-2A, where lysosomes can be observed inside the RPE, are provided in **Figure EV2**. Moreover, since methodologies available to study lysosomal exocytosis cannot be implemented in human donor RPE/eye samples, we have removed it from the text.

- Line 141: the role and interest of studying N-CoR1/RAR α ratio need to be introduced. We thank the reviewer for pointing this out and have now introduced N-CoR1/RAR α -mediated regulation of CMA in the text:

“We have previously described that in several tissues and cell types, including the retina, CMA is modulated at the transcriptional level by the RAR α transcription factor, and that the inhibitory effect of RAR α on CMA can be reversed by enhancing its interaction with the co-repressor N-CoR1.”

- Line 146: I do not understand why the authors are stating that they observed "a coetaneous impairment of CMA in the RPE...". Coetaneous with what? They might want to add a sentence that summarize all the deregulated processes seen in AMD patients samples.

We have rephrased this sentence and now also mention other cellular RPE alterations observed in donors, some of them partially or fully recapitulated in iPSC-RPE models.

- Line 156, in iPSC-derived RPE they observe decrease LAMP-2A proteins without changes in mRNA levels. They should comment on that as it is also different from what it is observed in patients samples?

While we did not observe a significant decrease of the *LAMP2A* splicing variant in iPSC-RPE cells, total amount of *LAMP2* transcripts did show a trend towards a decrease both when analyzed in the RPE cluster of scRNA-seq donor data (**Fig. Rev. 1a**) and by RT-qPCR of AMD iPSC-RPE cell lines (**Fig. Rev. 1b**; from **Figure EV4C**).

Figure for Reviewers 1. *LAMP2* mRNA levels in humans with AMD and donor-derived iPSC-RPE.

Our groups and others have previously described important contributions of posttranslational mechanisms to the regulation of levels of LAMP-2A in other cell types and organs. For example, the marked increase in LAMP-2A levels during starvation in liver does not require de novo protein synthesis and is mediated instead by reduced rates of LAMP-2A turnover in lysosomes (Cuervo & Dice, 2000). Conversely, in the context of aging, in many organs mRNA levels of LAMP-2A are still preserved, but levels of LAMP-2A protein decrease markedly due to aberrant degradation/reduced stability of LAMP-2A in lysosomes (Jafari *et al*, 2024; Kiffin *et al*, 2007). Considering the changes in lysosomal homeostasis associated with AMD it is very likely that the decrease in LAMP-2A protein in iPSC-derived RPE cells may be a consequence of reduced stability in this compartment. An alternative possibility is that translation of LAMP-2A is impaired in the context of AMD, or that as described in cystinosis patients (Zhang *et al*, 2019), trafficking of LAMP-2A from Golgi to lysosomes is reduced, leading to its degradation as part of Golgi quality control. On this respect, it is interesting to point out that levels of Rab11a, a key protein for LAMP-2A trafficking to lysosomes (Poehler *et al*, 2014), were markedly increased in patient derived cells upon treatment with the CMA activator (**Figure EV7B**). We have now included a brief version of this explanation in the text.

Globally, results would benefit from clarifications as it is not always very easy to follow all the hallmarks that are deregulated or not.

We have updated the text according to the reviewer's concerns and now also include a graphical abstract that summarizes the alterations observed.

Methods:

- Line 384: the protocol used for the obtention of iPSC and then the RPE should be re-written even if it is coming from a previous publication, with a separation between the obtention of iPSC and the differentiation of RPE cells.

We thank the reviewer for this suggestion; we have now described in depth the obtention and reprogramming of conjunctiva cells and separated this paragraph in "Donor-derived iPSC-RPE obtention" and "Donor-derived iPSC-RPE differentiation".

- Line 410: FFPE slides should be explained. Maybe the paragraphs of immunohistochemistry and immunocytochemistry could be reunited?

We have included additional information about FFPE obtention and processing prior to immunohistochemistry. Moreover, we have also joined together immunohistochemistry and immunocytochemistry sections.

- Line 461: the process of using lentiviruses should be explained.

We now provide a more in-depth protocol of lentiviral vector packaging in HEK293T cells and viral transduction into differentiated iPSC-RPE cells.

- Line 474: the term BGT should be explained.

BGT stands for BSA-Glycine-Triton buffer, we have now added this information to the methods section together with its exact composition (3 mg/mL BSA, 0.25% Triton X-100 mM Glycine in PBS).

Referee #2 (Comments on Novelty/Model System for Author):

The study provides novel insight on the role of a specific arm of autophagy pathways in dry age-related macular degeneration. While the findings are generally interesting, they could be improved by demonstration of selectivity of their tool compound CA77.1 as well as demonstration of clinical relevance in models of disease.

Referee #2 (Remarks for Author):

In the study "Defective chaperone-mediated autophagy in the retinal pigment epithelium of age-related macular degeneration patients" by Shang et al, the authors provide a series of data from human and conjunctiva-derived iPSC of RPE that demonstrates a general reduction of chaperone-mediated autophagy (CMA) (one of the 3 autophagy pathways that is specific for proteins containing KFERQ-like motifs) in patients with AMD. The authors further show that treatment with an activator of CMA reestablishes proteostasis and reestablishes RPE proteomes in AMD donor RPE. They propose that CMA activators can be explored therapeutically for AMD. The manuscript is well written, and study well executed. I have the following points for amelioration. Specifically with regards to the use of CMA stimulators and their clinical relevance.

We thank the reviewer for their feedback that has helped improve our manuscript, one of the main limitations of AMD research is the lack of appropriate animal models to understand the disease and test drugs for novel pharmacological targets. We now have further discussed it in the Discussion section.

-Figure 1e: The claim for extracellular vesicles in Drusen need substantiation. A higher magnification image would be desirable and evidence that the corresponding vesicles are indeed exocytosed. Drusen may have cells in them so a counter stain would help. We apologize for the low resolution of microscopy images as a result of file conversion during manuscript preparation. We counterstained cell nuclei with DAPI and did not observe any cells infiltrated within drusen. Since methodologies available to study lysosomal exocytosis cannot be reliably implemented in human donor RPE/eye samples, we have removed it from the text.

- I would suggest working in the table of patient baseline characteristics for the iPSC experiments, into the main manuscript.

We believe that the use of diverse patient-derived iPSC-RPE lines is one of the strengths of our study and thank the reviewer for this suggestion. Key information (sex, age, genotyping for common risk variants and MGS grade based on AREDS guidelines) has now been summarized in new **Table 1**. Additional ophthalmological observations and medical co-morbidities are provided in **Table EV4**.

- The authors describe that healthy iPSC-RPE showed preferential lysosomal 200 degradation of proteins involved in glycolysis and metabolic adaptation, while AMD 201 iPSC-RPE showed lysosomal degradation of proteins involved in fatty acid metabolism, 202 ferroptosis and ephrin signaling (Figure 3c). Can the authors elaborate here what this potential switch in effectors of energy metabolism means?

Previous research from our group has shown that the RPE of AMD donors presents a shift towards a more glycolytic metabolism. Proteomic analyses of RPE samples from

donors show an enrichment in glycolytic enzymes (Karunadharmaraj *et al*, 2022) already observed in MGS2/early AMD (Shen *et al*, 2023). Seahorse respirometry of primary hARPE and iPSC-RPE also showed decreased mitochondrial respiratory capacity in AMD samples (Fisher *et al*, 2022)(Ebeling *et al*, 2021). Additionally, studies by Golestaneh *et al*. have shown increased glycolysis-mediated ATP production in primary AMD hARPE (Golestaneh *et al*, 2017). Our results shed some light on this paradigm by including CMA in the equation: defective CMA-mediated degradation of glycolytic enzymes may favor and propel these metabolic alterations and can be corrected by turning up CMA (e.g. with CA77.1). Future studies will focus on studying the contribution of impaired CMA-mediated turnover of specific proteins to the AMD phenotype.

-The authors use CA77.1 and a concentration of 10uM. This is quite elevated. What data can the authors present on the specificity and pharmacodynamic properties of the compound given this elevated dose? An increase in NRF2 may actually be a stress response to the compound.

We initially described this family of CMA-activating compounds in (Gomez-Sintes *et al*, 2022). For most cell types analysed (NIH3T3, N2a, primary human fibroblasts), 10 μ M was the lowest dose able to significantly increase CMA and it did not have any effects on cell viability. Similarly, we did not observe any cell death or morphological alterations in healthy or AMD iPSC-RPE cells treated with 10 μ M CA77.1 for 24 hours. A full transcriptional analysis of treated and untreated cells, included in that study, confirmed absence of activation of stress-related pathways or cell death programs upon treatment.

Regarding the upregulation of NRF2, previous studies have demonstrated that CMA contributes to the antioxidant response in large part through the degradation of Keap1, the transcriptional repressor of NRF2, thereby leading to increased expression of NRF2 (Zhu *et al*, 2022). **Please see response to point 8) from Referee #3.**

- I found the labeling of figure 5 hard to follow with respect to what groups were compound stimulated. Please provide legends throughout.

We have updated the labelling of **Figures 4, 5** and **EV7**, indicating the iPSC-RPE donors (Healthy/AMD) and whether they were treated with CA77.1 (Veh/CA). We thank the reviewer for this suggestion and hope these figures are now clearer.

- Any attempts for in vivo validation of concepts presented in models of disease?

While we have not yet been able to validate our findings *in vivo* due to the lack of models that recapitulate the progression of AMD, this is now discussed along with other limitations of our study in the Discussion section. However, we have previously validated the neuroprotective effect of CA77.1 in other diseases. For example, in (Gomez-Sintes *et al*, 2022) we were able to slow down photoreceptor degeneration and preserve vision in the *rd10* model of retinitis pigmentosa by administering CA77.1 intraperitoneally (40 mg/kg) or intravitreally (40 μ M). In (Bourdenx *et al*, 2021), we showed that oral treatment with CA77.1 (30 mg/kg) in fronto-temporal dementia (PS19) and Alzheimer's disease (Tg.TauPS2APP) alleviated proteotoxicity, reduced neuroinflammation and improved cognitive function. We believe that CA77.1 holds a

huge therapeutic potential for AMD as well as other pathologies where proteotoxicity is involved in the etiology or progression of the disease.

Referee #3 (Comments on Novelty/Model System for Author):

Both models and methodologies are state-of-the-art the study may lead to novel pharmacological approaches in AMD.

Referee #3 (Remarks for Author):

Loss of autophagy pathways is linked to several, if not all, age-related diseases. In this manuscript, the authors report that the activity of chaperone-mediated autophagy (CMA) declines in the retinal pigment epithelium (RPE) in age-related macular degeneration patients, compared with healthy age-matched individuals. Previous studies had pointed out decreased macroautophagy, lysosomal and mitochondrial dysfunctions as potential causative factors in AMD, but whether LAMP-2A lysosomes specifically involved in CMA contributed to AMD pathology remained elusive. Using elegant orthogonal approaches in AMD donor samples and iPSC-RPE lines derived from donor's conjunctiva, the authors uncovered that defective CMA is a major contributor to RPE dysfunctions. Specifically, they found that CMA is responsible for the proteostasis of a selective subsets of KFERQ-containing proteins, involved in lipid metabolism among others pathways. Importantly, pharmacological treatment with a CMA activator rescued the affected proteome of the cells from AMD donors and AMD iPSC-RPE, an effect attributed to lowering oxidative stress, through the upregulation of NRF2, and improved mitochondrial function. Altogether, the manuscript presents some very interesting and clinically relevant data elucidating a mechanism by which CMA governs the healthy state of RPE and identifies CMA as an actionable target in AMD. The experimental design and the quality of the data are robust and support the conclusions of the study. A few points need to be further addressed by the authors to clarify some findings and/or further strengthen the main conclusions of the manuscript.

We thank the reviewer for their in-depth review and critical feedback that has helped strengthen our work.

Main comments:

1) Figure 1b, is the CMA score significantly reduced in the RPE of patients with AMD? Please be careful to describe results that do not reach statistical significance as a trend, and remove the word "significant/ly" from the descriptions of the data.2b

We have toned down our initial claims as the *P*-value did not reach significance (*P* = 0.0536). The text now reads:

"CMA score showed a trend towards a reduction selectively in the RPE (RPE65+BEST1+ cells) of patients with AMD..."

2) Under serum starvation CMA activity is increased (figure 2f), but most likely also macroautophagy is under these conditions. What is the relative contribution of these autophagy pathways to the AMD phenotype? Is the ProteoStat+ staining as a proxy of CMA, also rescued by starvation?

Previous results have described macroautophagy alterations in primary hARPE models and sections from AMD donors (Mitter *et al.*, 2014)(Golestaneh *et al.*, 2017). However, we have not found any differences in macroautophagy flux or lysosomal function in AMD iPSC-RPE compared to healthy iPSC-RPE (**Figure EV6**). Moreover, our KFERQ-Dendra reporter is specific for CMA since it is targeted to lysosomes by Hsc70 and “handled over” selectively to the LAMP-2A receptor for entrance of the substrate into the lysosome (**Figure 2E**). We have also extensively proven before that CA77.1 does not alter macroautophagy levels (Gomez-Sintes *et al.*, 2022), compared to previous CMA-activating compounds (Anguiano *et al.*, 2013). Therefore, we are confident that the defects observed on proteostasis and the phenotypic amelioration are caused by CMA down- and up-regulation, respectively.

3) Data on perturbed glutathione biosynthesis in AMD are interesting. Can the authors actually measure the levels of GSH in their iPSC-RPE lines? Please note that 4-hydroxynonenal (4-HNE) is a byproduct of lipid-peroxidation.

We thank the reviewer for this suggestion. Since we observed altered turnover of proteins involved in lipid peroxidation and ferroptosis, such as ACSL4, glutathione biosynthesis could also be dysregulated. We have not observed any differences in the levels of glutathione peroxidase 4 (GPX4) in iPSC-RPE (**Fig. Rev. 2a**), and a small trend towards an increase of reduced glutathione (GSSG) in culture medium (**Fig Rev. 2b**). Interestingly, we recently described an increase in GPX4 levels in the RPE of donors with early AMD (MGS2), which may indicate that this pathway may be altered once the RPE is within its own microenvironment and has to undertake lipid-rich photoreceptor OS recycling.

Figure for Reviewers 2. Changes in GPX4 and reduced glutathione levels in AMD iPSC-RPE.

4) Can the author explain in the text the selection criteria, based on the fold-change (FC >1.25), p-value ($p < 0.01$) to filter the number of enriched protein hits in healthy and AMD conditions?

Cut-offs were established based on previous experience from our group using proteomics data to assess KFERQ protein degradation through CMA (Bourdenx *et al.*, 2021; Shang *et al.*, 2024). This threshold was set considering the duration of the assay and the frequently long half-life of the proteins undergoing degradation through CMA. This information is now included in the methods section.

5) In the western blot of Figure 3e basal levels of LAMP-2A are not different from healthy and AMD derived iPSC-RPE? Also from the representative WB ACSL4 levels

are markedly elevated in basal conditions in AMD when compared with Healthy donors, but this is not reflected in the quantifications in panel 3f. Are there other ferroptosis-related proteins that accumulate as CMA declines in AMD, which could be contributing to the increased level of oxidative stress observed under this pathological condition? And related to this, have the authors observed ferroptosis as a contributing factor in the degeneration of AMD-derived iPSC-RPE cell lines?

In our hands, assessing proteostasis with iPSC-RPE lines originated from different donors can be difficult due to the differences in basal levels of most proteins, thus why we have performed paired analysis and relied on functional assays such as the KFERQ-Dendra or Seahorse. We have performed an unpaired Student's *t*-test to compare the basal levels of ACSL4 shown in **Figure 3F** but there were no significant differences ($P = 0.4384$). Additionally, we have replaced the western blot in **Figure 3E** with a more representative one of donors at the average of LAMP-2A protein levels for each group (Healthy: 1.0, AMD: 0.83).

Regarding ferroptosis, we would like to clarify that in AMD iPSC-RPE there are no signs of degeneration or increased levels of ferroptotic cell death but are rather a clean tool to assess the contributions of genetic makeup to cell death. In this sense, ongoing studies in the lab are focused on studying the contribution of impaired protein turnover of these new targets to the different pathways they are involved in, such as ferroptosis or lipid metabolism.

6) Can the authors discuss further how the lysosomal degradation of KFERQ-motif containing proteins affects cholesterol metabolism? Do the AMD-derived iPSC-RPE cells have problems in maintaining cholesterol fluxes (uptake and efflux)? Is cholesterol accumulated in the lysosomes of AMD-derived cells?

Previous studies from our groups and others, have shown that enzymes involved in lipogenesis as well as structural proteins that contribute to regulate lipolysis can become CMA in specific cell types and organs. For example in liver, CMA failure results in steatosis as result of expansion of number and size of lipid droplets (Schneider *et al*, 2014). In contrast, during adipogenesis the inability to reduced glycolysis upon CMA blockage, leads to reduced lipid droplet size (Kaushik *et al*, 2022). However, in neither of those instances, lysosomal membrane integrity or biology seemed to display any changes. Similarly, , we did not observe any differences in lysosomal function (**Figure EV5B**), acidity and membrane integrity (**Figure EV5C**) or morphology (**Figure EV5D**) in AMD-derived iPSC-RPE cells. While the iPSC-RPE model is useful to study the contribution of the genetic makeup of AMD donors to RPE cell biology (*e.g.* CMA-regulated proteostasis) they do not recapitulate some of the alterations seen in patients or haRPE cells such as lysosomal enlargement and lipofuscin accumulation. These alterations are due to the fact that the RPE must endure a lifetime of daily phagocytosis and degradation of photoreceptor OS. While this may be recapitulated in iPSC-RPE subjected to chronic OS feeding, these experiments are complex and we believe are out of the scope of the current manuscript. Nonetheless, we have discussed this and other potential limitations of the iPSC-RPE model in a new paragraph within the Discussion section.

7) In the schematic panel of Fig 3d the ULK1/2 inhibitor MRT68921 seems to block degradation of the autolysosome, whereas it prevents autophagosome formation. In

line with this it does not increase LC3-II accumulation as shown in WB of Fig. 3e (it should cause an accumulation of the LC3-I form).

We thank the reviewer for pointing this out, we have modified **Figure 3D** accordingly. Additionally, we have graphed the levels of LC3-I and LC3-II in our flux experiments to validate the inhibition of LC3 lipidation (MRT68921) or lysosomal degradation of LC3-II (NH₄Cl+Leupeptin) (**Fig. Rev. 3**). As anticipated by the reviewer, we do see an increase in LC3-I upon MRT treatment and an increase in LC3-II upon N/L treatment. Since we believe these validation is key for the assay, we have included these data as well in **Figure EV4F**.

Figure for Reviewers 3. Changes in LC3-I and LC3-II levels in iPSC-RPE treated with macroautophagy (MRT68921) or lysosomal degradation (Leupeptin+NH₄Cl) inhibitors.

8) Data using the CMA activator is captivating. Mechanistically does CA77.1 increase the expression levels and nuclear translocation of NRF2 in AMD-derived RPE by targeting Keap1? Is the enhanced NRF2 activity responsible for LAMP-2A expression after CA77.1 treatment? This could indicate a feedforward positive loop. Please also consider to increase the quality of Fig 4b, nuclear immunostaining of NRF2 is barely detectable.

We apologize for the low resolution of microscopy images as a result of file conversion during manuscript preparation. Additionally, we have linearly increased the contrast and brightness, increased the magnification and now only show the DAPI outline in **Figure 5B**, we hope it better illustrates the differences in nuclear/cytosolic NRF2. Indeed Keap1 has previously been described as a substrate for CMA (Zhu *et al.*, 2022) and its accumulation in AMD iPSC-RPE could explain the decreased levels of nuclear NRF2 (**Fig. 5B**). We have now included this possibility in the discussion.

We have performed additional western blot flux experiments, but we were not able to detect any differences in basal Keap1 protein levels (**Fig. Rev. 4A**), nor validate Keap1 as a CMA substrate in iPSC-RPE since it did not increase in N/L-treated cells irrespective of AMD status (**Fig. Rev. 4B**). Moreover, its levels did not change when we treated cells with CA77.1 (**Fig. Rev. 4C**). Nonetheless, since its turnover has previously been linked to CMA it may also be context-dependent (e.g. in response to additional stressors) in the case of iPSC-RPE.

On the other hand, *LAMP2* mRNA levels have also been described to be regulated by NRF2 (Pajares *et al*, 2018). Accordingly, we did observe an increase in *LAMP2* mRNA levels in both healthy and AMD iPSC-RPE (Fig. Rev. 4D). However, we cannot rule out that this is due to a direct effect of decreased RAR activity since we have previously observed a similar increase in *LAMP2* mRNA levels without changes in *Nfe2i2/NRF2* in other cell types (Gomez-Sintes *et al.*, 2022). We have now added the possibility of a feedforward positive loop, as proposed by the reviewer, in the discussion.

Figure for Reviewers 4. Keap1 levels are not regulated by CMA in iPSC-RPE.

References

- Anguiano J, Garner TP, Mahalingam M, Das BC, Gavathiotis E, Cuervo AM (2013) Chemical modulation of chaperone-mediated autophagy by retinoic acid derivatives. *Nat Chem Biol* 9: 374-382
- Bourdenx M, Martin-Segura A, Scrivo A, Rodriguez-Navarro JA, Kaushik S, Tasset I, Diaz A, Storm NJ, Xin Q, Juste YR *et al* (2021) Chaperone-mediated autophagy prevents collapse of the neuronal metastable proteome. *Cell* 184: 2696-2714 e2625
- Cuervo AM, Dice JF (2000) Regulation of lamp2a levels in the lysosomal membrane. *Traffic* 1: 570-583
- Ebeling MC, Geng Z, Kapphahn RJ, Roehrich H, Montezuma SR, Dutton JR, Ferrington DA (2021) Impaired Mitochondrial Function in iPSC-Retinal Pigment Epithelium with the Complement Factor H Polymorphism for Age-Related Macular Degeneration. *Cells* 10
- Fisher CR, Ebeling MC, Geng Z, Kapphahn RJ, Roehrich H, Montezuma SR, Dutton JR, Ferrington DA (2022) Human iPSC- and Primary-Retinal Pigment Epithelial Cells for Modeling Age-Related Macular Degeneration. *Antioxidants (Basel)* 11
- Golestaneh N, Chu Y, Xiao YY, Stoleru GL, Theos AC (2017) Dysfunctional autophagy in RPE, a contributing factor in age-related macular degeneration. *Cell Death Dis* 8: e2537
- Gomez-Sintes R, Xin Q, Jimenez-Loygorri JI, McCabe M, Diaz A, Garner TP, Cotto-Rios XM, Wu Y, Dong S, Reynolds CA *et al* (2022) Targeting retinoic acid receptor alpha-corepressor interaction activates chaperone-mediated autophagy and protects against retinal degeneration. *Nature communications* 13: 4220
- Jafari M, Macho-González A, Diaz A, Lindenau K, Santiago-Fernández O, Zeng M, Massey AC, de Cabo R, Kaushik S, Cuervo AM (2024) Calorie restriction and calorie-restriction mimetics activate chaperone-mediated autophagy. *Proceedings of the National Academy of Sciences of the United States of America* 121: e2317945121
- Karunadharma PP, Kapphahn RJ, Stahl MR, Olsen TW, Ferrington DA (2022) Dissecting Regulators of Aging and Age-Related Macular Degeneration in the Retinal Pigment Epithelium. *Oxid Med Cell Longev* 2022: 6009787
- Kaushik S, Juste YR, Lindenau K, Dong S, Macho-Gonzalez A, Santiago-Fernandez O, McCabe M, Singh R, Gavathiotis E, Cuervo AM (2022) Chaperone-mediated autophagy regulates adipocyte differentiation. *Sci Adv* 8: eabq2733
- Kiffin R, Kaushik S, Zeng M, Bandyopadhyay U, Zhang C, Massey AC, Martinez-Vicente M, Cuervo AM (2007) Altered dynamics of the lysosomal receptor for chaperone-mediated autophagy with age. *J Cell Sci* 120: 782-791
- Mitter SK, Song C, Qi X, Mao H, Rao H, Akin D, Lewin A, Grant M, Dunn W, Jr., Ding J *et al* (2014) Dysregulated autophagy in the RPE is associated with increased susceptibility to oxidative stress and AMD. *Autophagy* 10: 1989-2005
- Pajares M, Rojo AI, Arias E, Díaz-Carretero A, Cuervo AM, Cuadrado A (2018) Transcription factor NFE2L2/NRF2 modulates chaperone-mediated autophagy through the regulation of LAMP2A. *Autophagy* 14: 1310-1322
- Poehler AM, Xiang W, Spitzer P, May VE, Meixner H, Rockenstein E, Chutna O, Outeiro TF, Winkler J, Masliah E *et al* (2014) Autophagy modulates SNCA/ α -synuclein release, thereby generating a hostile microenvironment. *Autophagy* 10: 2171-2192
- Schneider JL, Suh Y, Cuervo AM (2014) Deficient chaperone-mediated autophagy in liver leads to metabolic dysregulation. *Cell Metab* 20: 417-432
- Shang P, Ambrosino H, Hoang J, Geng Z, Zhu X, Shen S, Eminhizer M, Hong E, Zhang M, Qu J *et al* (2024) The Complement Factor H (Y402H) risk polymorphism for age-related macular degeneration affects metabolism and response to oxidative stress in the retinal pigment epithelium. *Free Radic Biol Med* 225: 833-845
- Shen S, Kapphahn RJ, Zhang M, Qian S, Montezuma SR, Shang P, Ferrington DA, Qu J (2023) Quantitative Proteomics of Human Retinal Pigment Epithelium Reveals Key Regulators for the Pathogenesis of Age-Related Macular Degeneration. *Int J Mol Sci* 24

Zhang J, He J, Johnson JL, Rahman F, Gavathiotis E, Cuervo AM, Catz SD (2019) Chaperone-Mediated Autophagy Upregulation Rescues Megalin Expression and Localization in Cystinotic Proximal Tubule Cells. *Front Endocrinol (Lausanne)* 10: 21

Zhu L, He S, Huang L, Ren D, Nie T, Tao K, Xia L, Lu F, Mao Z, Yang Q (2022) Chaperone-mediated autophagy degrades Keap1 and promotes Nrf2-mediated antioxidative response. *Aging Cell* 21: e13616

18th Sep 2025

Dear Dr. Jiménez-Loygorri,

Thank you for submitting your revised study. We have now received the reports from the referees. As you will see below, they are satisfied with the revisions, and I will therefore be able to accept your manuscript once the following editorial concerns are addressed:

1/ Manuscript text:

- Please remove the blue font text and only keep in track changes any new modification in the text.
- An email bounced for Raquel Gomez Sintes (rgomez@cib.csic.es), please check and correct if needed.
- Please correct the order and heading of the manuscript sections to: Abstract / Keywords / The Paper Explained / Introduction / Results / Discussion / Methods / Data Availability / Acknowledgements / Disclosure and Competing Interests Statement / References / Figure Legends / Tables / Expanded View Figure Legends.
- Methods:
 - o Please remove the Reagents and Tools Table from the manuscript text upload as a separate file; please update the EV table citations.
 - o Please provide a full statement confirming that informed consent was obtained from all subjects and that the experiments conformed to the principles set out in the WMA Declaration of Helsinki and the Department of Health and Human Services Belmont Report. Please state details of authority granting ethics approval (IRB or equivalent committee(s), provide reference number for approval.
 - o Please provide the origin of the cells, and indicate whether they were authenticated and tested for mycoplasma contamination.
- Data availability section: please provide a direct link to the deposited dataset. Please note that the datasets must be public before acceptance of the manuscript. Remove "Source data are provided with this paper".
- The information provided in the Acknowledgements should match the information provided in the submission system. Check whether the JPB Foundation and Hevolution Foundation need to be added as funders in the submission system. Please remove the BioRender information from this section, and add it in a dedicated section to the Methods following this format:
Graphics:
(some of the... OR Figure #... OR synopsis) Graphics were created with BioRender.com.

2/ Figures:

- Tables EV1 and 2 should be renamed Dataset EV1 and Dataset EV2. Both need a legend added to the corresponding file, in a separate tab/worksheet. The remaining EV tables will need to be renumbered and they all need a short legend added to the top of the page.
- Please correct the nomenclature to "Figure EV1" etc. and Table "EV1" etc. in the legends and in the table/figure files.
- Please make sure all figures/figure panels are referenced in the manuscript text, and in chronological order (currently, callouts are missing for Table 1 and Fig. 2E; Fig. EV3A is called out before Fig. EV2 and Table EV1 is called out after Table EV2, 3 and 4).
- Please address the queries from our data editors in the figure legends:
Please note that information related to n is missing in the legends of figures 1E, 2B, D, F, G, K; 3F, 4A, B, C, F, G, H; EV3 E, F; EV4 B-F; EV5 B-D; EV6 A, B, C, E, F, G, H, I, J, K, M; EV7 C

3/ In the checklist, please fill in:

- DNA and RNA sequences
- Cell materials/cell lines
- Experimental study design/randomization

4/ Please note that all corresponding authors are required to supply an ORCID ID for their name upon submission of a revised manuscript. An ORCID ID is missing for Ferrington.

5/ I introduced minor edits in your Paper Explained section, please let me know if you agree with the following, or amend as you see fit:

Problem:

Age-related macular degeneration (AMD) is the leading cause of irreversible central vision loss in elderly people, affecting the macula. As global populations age, the prevalence of AMD is projected to double by 2040. In AMD, the retinal pigment epithelium (RPE) degenerates early on, yet the molecular basis of its selective vulnerability remains poorly understood. We investigated a less well-explored cellular clearance pathway - chaperone-mediated autophagy (CMA) - and its potential role in RPE degeneration in AMD.

Results

We found that CMA activity is significantly reduced in the RPE of AMD donor eyes, leading to the accumulation of proteins containing KFERQ-like motifs-specific tags for CMA-mediated degradation. Using AMD patient-derived iPSC-RPE, we confirmed that these cells display impaired CMA, chronic proteotoxic stress and oxidative damage. Proteomic analysis revealed that both healthy and AMD iPSC-RPE cells rely on CMA to degrade proteins involved in metabolic regulation and signalling. Treatment with the pharmacological CMA activator CA77.1 increased LAMP2A expression and restored CMA activity. This reduced toxic protein buildup and improved the cellular energy balance of AMD iPSC-RPE.

Impact

Our findings reveal that the downregulation of CMA is a novel, RPE-specific defect in AMD that contributes to the loss of proteostasis and cellular resilience during disease progression. Pharmacological activation of CMA alleviated multiple hallmarks of RPE degeneration in patient-derived cells. These results open new therapeutic avenues for treating AMD by targeting CMA to enhance cellular clearance, restore metabolic function and protect the RPE. Modulating this pathway could be a strategy for preventing or slowing vision loss in AMD and possibly other retinal degenerations.

6/ I introduced minor changes in your synopsis, please let me know if you agree with the following or amend as you see fit: "Impaired CMA was found to be a specific vulnerability of the RPE during AMD. Pharmacological upregulation of CMA restored proteostasis and promoted metabolic fitness, highlighting the therapeutic potential of targeting CMA in AMD.

- CMA was impaired in the RPE of patients affected by AMD
- iPSC-RPE derived from donors with AMD recapitulated CMA impairment.
- CMA regulated different subsets of the proteome in healthy and AMD iPSC-RPE.
- Pharmacological activation of CMA restored proteostasis and promoted metabolic fitness.
- CMA activation stimulated NRF2-mediated antioxidant response and alleviated oxidative damage."

Thank you for providing a nice visual abstract. Please upload it as a png/tiff/jpeg file 550 px wide x 300-600 px high and make sure that the text remains legible. A cropped portion of this image will serve as thumbnail for the table of content on our webpage.

7/ As part of the EMBO Publications transparent editorial process initiative (see our Editorial at <http://embomolmed.embopress.org/content/2/9/329>), EMBO Molecular Medicine will publish online a Review Process File (RPF) to accompany accepted manuscripts.

This file will be published in conjunction with your paper and will include the anonymous referee reports, your point-by-point response and all pertinent correspondence relating to the manuscript. Let us know whether you agree with the publication of the RPF and as here, if you want to remove or not any figures from it prior to publication.

I look forward to receiving your revised manuscript.

Yours sincerely,

Lise Roth

***** Reviewer's comments *****

Referee #1 (Comments on Novelty/Model System for Author):

paper of interest for the community.

Referee #1 (Remarks for Author):

Authors have made most of the corrections asked and clarified greatly their paper. It is now suitable for publication.

Referee #2 (Comments on Novelty/Model System for Author):

The authors answered my questions by providing novel text and tables. The one point is that no additional experiments were performed on selectivity of their compound. This would have been performed if there is interest to bring the concepts forward.

Referee #3 (Remarks for Author):

The authors have improved the clarity and quality of the study and answered satisfactorily to this Reviewer's questions.

The authors addressed the remaining editorial issues.

13th Oct 2025

Dear Dr. Jiménez-Loygorri,

Thank you for submitting your revised files. I am pleased to inform you that your manuscript is accepted for publication and is now being sent to our publisher to be included in the next available issue of EMBO Molecular Medicine.

Yours sincerely,
